# Generalized Probabilistic Attention Mechanism in Transformers

## Abstract

The Transformer architecture has become widely adopted due to its demonstrated success, attributed to the attention mechanism at its core. Despite these successes, the attention mechanism of Transformers is associated with two well-known issues: rank-collapse and gradient vanishing. In this paper, we present a theoretical analysis that it is inherently difficult to address both issues simultaneously in the conventional attention mechanism. To handle these issues, we introduce a novel class of attention mechanism, referred to as generalized probabilistic attention mechanism (GPAM), and its dual-attention implementation within the Transformer architecture. Unlike conventional attention mechanisms, GPAM allows for negative attention scores while preserving a fixed total sum. We provide theoretical evidence that the proposed dual-attention GPAM (daGPAM) effectively mitigates both the rank-collapse and gradient vanishing issues which are difficult to resolve simultaneously with the conventional attention mechanisms. Furthermore, we empirically validate this theoretical evidence, demonstrating the superiority of daGPAM compared to other alternative attention mechanisms that were proposed to address the same issues. Additionally, we demonstrate the practical benefits of GPAM in natural language processing tasks, such as language modeling and neural machine translation.

## 1 Introduction

The Transformer model, as introduced by (Vaswani, 2017), has emerged as a pivotal architecture driving the advancement of contemporary deep learning models across various domains, including natural language processing (Brown et al., 2020), audio signal processing (Gulati et al., 2020), and image processing (Dosovitskiy et al., 2021). Central to the Transformer's success is the attention mechanism, which facilitates the contextualization of input token representations. Based on the similarities of query and key vectors, the attention mechanism mixes value vectors as follows (a single scaled dot-product self-attention head (Vaswani, 2017)):

$$\mathbf{Y} = \mathbf{P}\mathbf{X}_V, \tag{1}$$

$$\mathbf{P}_{ij} = \frac{exp(\mathbf{A}_{ij})}{\sum_{k=1}^{T} exp(\mathbf{A}_{ik})}, \tag{2}$$

$$\mathbf{A} = \left(\sqrt{d_{qk}}\right)^{-1} \mathbf{X}_Q \mathbf{X}_K^\top, \tag{3}$$

where $\mathbf{X}_Q = \mathbf{X}\mathbf{W}_Q$, $\mathbf{X}_K = \mathbf{X}\mathbf{W}_K$, and $\mathbf{X}_V = \mathbf{X}\mathbf{W}_V$. $\mathbf{X} \in \mathbb{R}^{T \times d}$ is input representations with $T$ sequence length and $d$ dimensionality, and $\mathbf{W}_Q, \mathbf{W}_K \in \mathbb{R}^{d \times d_{qk}}$, and $\mathbf{W}_V \in \mathbb{R}^{d \times d_v}$ are weight matrices.

Despite its empirical success, the conventional attention mechanism, particularly the self-attention mechanism, has been shown to exhibit two significant limitations. The first issue is the phenomenon known as the rank-collapse problem (Dong et al., 2021), where output token representations become similar to others, leading to a loss of rank as they progress through each layer. While a controlled degree of rank-reduction can be beneficial for eliminating redundant information (Tishby & Zaslavsky, 2015), excessive rank-reduction risks the loss of critical information. Previous studies have extensively documented the self-attention layer's tendency for intensive rank-reduction (Dong et al.,

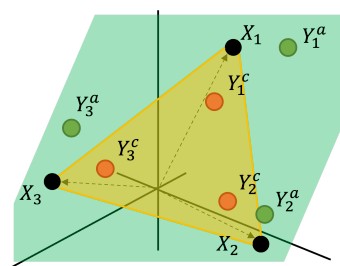
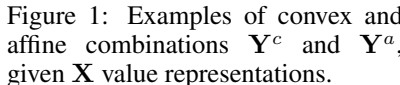

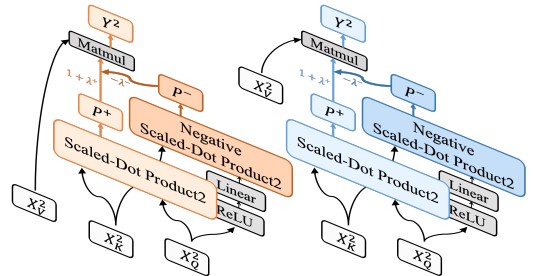

Figure 1: Examples of convex and affine combinations $\mathbf{Y}^c$ and $\mathbf{Y}^a$, given $\mathbf{X}$ value representations.

Figure 2: Our proposed daGPAM structure in an example of two multi-head self-attention in Transformer.

2021; Noci et al., 2022; 2024). The second issue is the gradient vanishing problem (Richter & Wattenhofer, 2022; Wang et al., 2021). During the softmax-based normalization step, the gradient is consistently less than 1 and saturates close to 0, thereby impeding upper attention layers' sufficient flow of gradients to the lower layers. Moreover, in this paper, we demonstrate that these two issues are inherently challenging to address both simultaneously.

Previous approaches to addressing the rank-collapse problem in Transformers have primarily focused on preserving input token representations. One strategy amplifies the coefficient of the short-cut branch in the residual connection (Noci et al., 2022), while another reduces the contextualization effect by regularizing the attention matrix, $\mathbf{P}$, to be similar to an identity matrix (He et al., 2023; Noci et al., 2024). However, these methods can diminish the attention layer's ability to capture meaningful contextual information (Veit et al., 2016; Zhang et al., 2022). To address the gradient vanishing problem, alternative attention mechanisms have been proposed to replace the conventional softmax-based mechanism, Eqs.2 and 3 (Richter & Wattenhofer, 2022; Wang et al., 2021). Despite their theoretical effects for improving gradient flow, these alternative mechanisms have not demonstrated practical advantages in benchmark experiments.

In contrast to prior approaches, we posit that the underlying issues stem from the convex combination structure in the conventional attention mechanism. The normalized attention scores, $\mathbf{P}_{ij}$, as defined in Eq.2, are non-negative and sum to 1 along the query axis, making the conventional attention mechanism a valid convex combination of input representations. Consequently, as illustrated in Fig.1, the output representations, $\mathbf{Y}^c$, are constrained within the convex hull (yellow plane) of the input value representations. This fundamental constraint limits the diversity of output representations, causing them to become more similar to one another than the input value representations, thereby intensifying the rank-collapse problem.

This consideration naturally raised the question: how about making it an affine combination? The affine combination is a generalization of the convex combination, allowing the normalized attention scores to take negative values while still maintaining a total sum to be 1. As depicted in Fig.1, the output representations, $\mathbf{Y}^a$, are no longer constrained to lie within the convex hull, but instead within the affine hull (green plane), thereby removing the fundamental constraint that can lead to rank-collapse. However, from a probabilistic perspective, the notion of negative normalized attention scores may seem unconventional, since negative probability is unfamiliar concept. Nevertheless, the discussions by prominent physicists, such as P. Dirac (Dirac, 1942) and R. Feynman (Feynman, 1984), have concretized the concept of negative probability by generalizing the Kolmogorov's probability conditions as follows:

**Kolmogorov's Probability Conditions..** *(1) $P(e) \geq 0$, and (2) $\sum\limits_{e \in \Omega} P(e) = 1$ where $P(e) \in \mathbb{R}$ is the probability measure of event $e$ and $\Omega$ is event space.*[1]

**Generalized Probability Conditions (Székely, 2005).** *(1) $\sum\limits_{e \in \Omega} |P(e)| < \infty$, and (2) $\sum\limits_{e \in \Omega} P(e) = 1$ where $P(e) \in \mathbb{R}$ is the probability measure of event $e$ and $\Omega$ is event space.*

---

[1]For simplicity, we mention only the two main conditions.

It is important to note that the first condition of generalized probability allows for the possibility of negative probabilities. Building on this long-standing line of research, we introduce the attention mechanism based on the affine (or scaled-affine) combination, which we refer to as the generalized probabilistic attention mechanism (GPAM). As a generalized framework, GPAM encompasses the conventional attention mechanism as a special case where all attention values are non-negative.

In this paper, we explore GPAM in the Transformer architecture, and as a cornerstone design of GPAM, we propose a dual-attention design that facilitates GPAM with adding only a negligible number of parameters. Specifically, we add an additional attention matrix computation to the original scaled dot-product self-(or cross-) attention mechanism with a small additional weight matrix which increases less than 1% of total parameters on average. Then, the resulting two attention matrices are treated as positive and negative parts of the final attention score, respectively. By combining these two attention scores with pre-defined or trainable scalar weights, we ensure that our proposed dual-attention GPAM (daGPAM) a valid affine (or scaled-affine) combination. Fig.2 illustrates the structure of this design. When we propose a GPAM method, we show theoretically that our method is advantageous for mitigating not only the rank-collapse problem, but also the gradient vanishing problem. Our empirical validations give evidences for the above theories in various aspects. In addition, we experimentally demonstrate and explain the superiority of daGPAM compared to other alternative attention mechanisms that was proposed to address the mentioned problems. Finally, we demonstrate the benefits of daGPAM in benchmark experiments in tasks such as language modeling (LM) and neural machine translation (NMT).

## 2 RELATED WORKS

### 2.1 RANK-COLLAPSE PROBLEM IN TRANSFORMER

Recent studies have analyzed the occurrence and practical risks of the rank-collapse problem in Transformers (Dong et al., 2021; Yan et al., 2022; Noci et al., 2022; He et al., 2023; Noci et al., 2024). The rank-collapse phenomenon refers to the tendency of token representations to become increasingly similar to one another as they are processed through successive layers. The first theoretical exploration of this issue demonstrated that the pure attention layer, as described by Eqs.1∼3, reduces the '*residual*' at an exponential rate with increasing layer depth. The *residual* metric quantifies how close token representations are to their mean point in terms of Euclidean distance, and is formally defined as follows:

$$res(\mathbf{X}) = \mathbf{X} - \mathbf{1}\bar{\mathbf{x}}^\top \in \mathbb{R}^{T \times d}, \text{ where } \bar{\mathbf{x}} = \arg\min_{\mathbf{x}} \|\mathbf{X} - \mathbf{1}\mathbf{x}^\top\|. \tag{4}$$

Then, the relationship of input/output *residual* is derived as follows:

**Lemma 1** (Dong et al. (2021), Simplified). *For any single scaled dot-product self-attention layer with a term $\gamma$ that depends on the attention entries, the composite norm of output residual is bounded by*

$$\|res(\mathbf{Y})\|_{1,\infty} \leq \frac{4\sqrt{2}\gamma \|\mathbf{W}_{QK}\|_1 \|\mathbf{W}_V\|_{1,\infty}}{\sqrt{d_{qk}}} \|res(\mathbf{X})\|_{1,\infty}^3, \tag{5}$$

*where $\mathbf{W}_{QK} = \mathbf{W}_Q \mathbf{W}_K^\top$. In the region that holds $4\sqrt{2}\gamma \|\mathbf{W}_{QK}\|_1 \|\mathbf{W}_V\|_{1,\infty} < \sqrt{d_{qk}}$, the output residual norm is diminished compared to the cubic rate of input residual norm.*

Building on this lemma and extending it to multi-layer cases, prior research has argued for the existence of the rank-collapse problem and empirically demonstrated that a substantial region of the parameter space that falls into rank-collapse issue (Dong et al., 2021). For a complete description of the lemma, please refer to Appendix A.1.1, and read the original paper for the proof. In addition to the rank-collapse problem, related concepts such as attention collapse, over-smoothing, over-correlation, and dimensional collapse have been identified in various domains, including vision Transformers (Zhou et al., 2021; Tang et al., 2021; Gong et al., 2021; Wang et al., 2022), contrastive learning (Jing et al., 2021; Hua et al., 2021), graph neural networks (Li et al., 2018; Jin et al., 2022; Guo et al., 2023; Roth & Liebig, 2024), and general neural networks (Feng et al., 2022; Jacot, 2023).

Previous studies have attempted to address the rank-collapse problem through various strategies, such as strengthening the shortcut branch (Tang et al., 2021; Noci et al., 2022), regularizing the attention matrix to be similar to an identity matrix (He et al., 2023; Noci et al., 2024), and regularizing

to preserve local information (Yan et al., 2022). However, these approaches may weaken the contextualization effect of the attention layer. Alternative methods have proposed various sequence-wise normalization techniques aimed at explicitly diversifying token representations (Hua et al., 2021; Guo et al., 2023). However, these techniques may be unsuitable for short sentences and autoregressive architectures due to their reliance on inadequate statistic across sequences. In contrast to these approaches, our GPAM develops the attention layer using a novel methodology grounded in the principles of convex and affine combinations.

## 2.2 Gradient Vanishing Problem in Transformer

Several previous studies have identified the gradient vanishing problem within the Transformer architecture (Zhang et al., 2019; Richter & Wattenhofer, 2022; Liu et al., 2020; Wang et al., 2021). A primary contributor to this issue is the gradients of the softmax function, as represented in Eq.2, which consistently yield gradient values less than 1. The following lemma derives these gradients.

**Lemma 2** (Gradient Vanishing in Attention Mechanism). *The gradient that the input unnormalized attention score, Eq.3, receives through the normalized attention score, Eq.2, (without $1/\sqrt{d_{qk}}$) is derived as follows:*

$$\frac{\partial \mathbf{P}_{ij}}{\partial \mathbf{A}_{ij}} = \mathbf{P}_{ij}(1 - \mathbf{P}_{ij}), \quad \frac{\partial \mathbf{P}_{ik,k \neq j}}{\partial \mathbf{A}_{ij}} = -\mathbf{P}_{ik}\mathbf{P}_{ij}, \tag{6}$$

*The maximum magnitude of each gradient is 0.25 when $\mathbf{P}_{ij} = 0.5$ for the first, and $\mathbf{P}_{ij} = 0.5$ and $\mathbf{P}_{ik} = 0.5$ for the last.*

Previous research has demonstrated that there exists a significant saturation range of $\mathbf{A}_{ij}$ values, leading to the gradients in Eq.6 approaching close to 0 (Richter & Wattenhofer, 2022).

Previous studies have sought to address the gradient vanishing problem by proposing various forms of unnormalized attention scores, including non-exponentiated raw scores (Richter & Wattenhofer, 2022) and scores transformed by periodic functions (Wang et al., 2021). Additionally, some approaches have eliminated the normalization step entirely to ensure that the gradient remains equal to 1 (Richter & Wattenhofer, 2022). While these approaches are theoretically effective in mitigating the gradient vanishing issue, they have not demonstrated practical advantages in benchmark experiments.

## 2.3 Alternative Attention Mechanisms

Among the previous studies that have proposed alternative attention mechanisms comparable to our GPAM (Wang et al., 2021; Richter & Wattenhofer, 2022; He et al., 2023; Noci et al., 2024), some methods unintentionally permit negative attention scores, similar to our approach. Additionally, there is prior research that intentionally incorporates negative attention scores within its framework (Tay et al., 2019). However, most of these studies do not adhere to the generalized probability conditions, specifically: (1) a finite range and (2) a total sum equal to 1 (or another fixed value). We will discuss the significance of adhering to these conditions and demonstrate the practical advantages through experiment results.

## 3 Relationship of Rank-Collapse and Gradient Vanishing

In this section, we analyze an inherent relationship between the rank-collapse and gradient vanishing problems within the conventional attention mechanism. We conjecture that the maximum total norm of gradients, defined as $G(\mathbf{P}_i) = \sum_{j=1}^{T} \|\frac{\partial \mathbf{P}_{ik}}{\partial \mathbf{A}_{ij}}\|_1$, is attained when complete rank-collapse occurs. We substantiate this conjecture through the following lemma.

**Lemma 3** (Maximum Total Norm of Gradients). *The total norm of gradients, $G(\mathbf{P}_i)$, is maximized when $\mathbf{P}_i$ is the uniform distribution, that is $\mathbf{P}_i = [\frac{1}{T}, \frac{1}{T}, \cdots, \frac{1}{T}]$ which is the case of complete rank-collapse.*

See Appendix A.1.2 for the proof. To mitigate the gradient vanishing problem, it is better to input similar token representations to make a smooth normalized attention score distribution. However, that remedy can make the rank-collapse problem severe. Therefore, mitigating both problems together is challenging.

## 4 DUAL-ATTENTION STRUCTURE OF GENERALIZED PROBABILISTIC ATTENTION MECHANISM

In this section, we explain our proposed dual-attention implementation of GPAM within the Transformer architecture, as illustrated in Fig.2. We further elucidate the dynamics of daGPAM with respect to the output space and representations. Additionally, we present theoretical foundations supporting the assertion that our daGPAM effectively addresses both the rank-collapse and gradient vanishing problems simultaneously.

### 4.1 DUAL-ATTENTION GPAM (DAGPAM) STRUCTURE

Based on the original scaled dot-product attention mechanism (Vaswani, 2017), we add another process of computing negative attention matrix while the original attention matrix is treated as the positive attention matrix. During the computation of the negative attention matrix, we use different query vectors, but transformed from the original query vectors. Subsequently, both the positive and negative attention matrices are integrated to derive the final attention matrix, and it is multiplied to the value representations to output the final representations. This process is formulated as follows (we use '+' notation to the parts of the original attention part):

$$\mathbf{Y} = \mathbf{P}^G \mathbf{X}_V, \tag{7}$$

$$\mathbf{P}^G = (1 + \lambda^+)\mathbf{P}^+ - \lambda^- \mathbf{P}^-, \tag{8}$$

$$\mathbf{P}^+_{ij} = \frac{exp(\mathbf{A}^+_{ij})}{\sum_{k=1}^T exp(\mathbf{A}^+_{ik})}, \quad \mathbf{A}^+ = \left(\sqrt{d_{qk}}\right)^{-1} \mathbf{X}^+_Q \left(\mathbf{X}^+_K\right)^\top, \tag{9}$$

$$\mathbf{P}^-_{ij} = \frac{exp(\mathbf{A}^-_{ij})}{\sum_{k=1}^T exp(\mathbf{A}^-_{ik})}, \quad \mathbf{A}^- = \left(\sqrt{d_{qk}}\right)^{-1} \left(\sigma(\mathbf{X}^+_Q)\mathbf{W}^-_Q\right) \left(\mathbf{X}^+_K\right)^\top, \tag{10}$$

where $\mathbf{X}^+_Q = \mathbf{X}\mathbf{W}^+_Q$, $\mathbf{X}^+_K = \mathbf{X}\mathbf{W}^+_K$, and $\mathbf{X}_V = \mathbf{X}\mathbf{W}_V$. For the new, negative parts, we only add the non-linear activation, $\sigma$ (we used ReLU throughout this work), and linear transformation with weight matrix $\mathbf{W}^-_Q \in \mathbb{R}^{d_{qk} \times d_{qk}}$ that has small number of parameters (note that $d > d_{qk}$). $\lambda^+$ and $\lambda^-$ are pre-defined or trainable scalars that control the effect of positive and negative normalized attention scores, respectively.

daGPAM exhibits several notable properties concerning its final normalized attention scores, denoted as $\mathbf{P}^G$. Specifically, the total sum of these scores is $\Sigma = (1 + \lambda^+ - \lambda^-)^2$ and the range of the scores is constrained such that $-\lambda^- \leq \mathbf{P}^G_{ij} \leq (1 + \lambda^+)$. In the special case where $\lambda^+ = \lambda^- = 0$, daGPAM becomes the conventional attention mechanism, Eqs.1∼3. In general, when $\lambda^+ = \lambda^-$, therefore $\Sigma = 1$, daGPAM facilitates a valid affine combination. In the next section, we will further elucidate the case where $\lambda^+ \neq \lambda^-$, that facilitates scaled-affine combination whose affine hull is simply translated.

### 4.2 DYNAMICS OF DUAL-ATTENTION GPAM

In this section, we explain the dynamic how the output space and representations of daGPAM are influenced by the combination of $\lambda^+$ and $\lambda^-$. We begin by explaining the dynamic of the output space. Based on the formulations in the previous section, the final output representations can be derived and expressed as follows: $\mathbf{Y} = \Sigma(\hat{\mathbf{P}}^G \mathbf{X}_V) = \Sigma \hat{\mathbf{Y}}$ where $\hat{\mathbf{P}}^G_{ij} = \mathbf{P}^G_{ij}/\Sigma$. It is important to note that $\hat{\mathbf{Y}}$ represents a valid affine combination of $\mathbf{X}_V$ since the total sum of $\hat{\mathbf{P}}^G$ is equal to 1. Consequently, the possible outcomes for $\mathbf{Y}$ can be interpreted as $\Sigma$-scaled versions of the outcomes from $\hat{\mathbf{Y}}$ which reside within the affine hull defined by $\mathbf{X}_V$. This concept is visually represented in Fig.3, where each colored plane illustrates the translated affine hull corresponding to different combinations of $\lambda$s. It is noteworthy that the degree of translation is contingent upon $\Sigma$ rather than the specific values of $\lambda$s, so in this example, only $\lambda^-$ was adjusted. Constraining the output to lie within affine hull (or translated) is critical for effective information processing. The translated affine hull represents a lower-dimensional hyperplane, which is a condensed subset of the $d$-dimensional

---

[2]For simplicity, we utilize the $\Sigma$ symbol to represent $\sum_{j=1}^T \mathbf{P}^G_{ij}$.

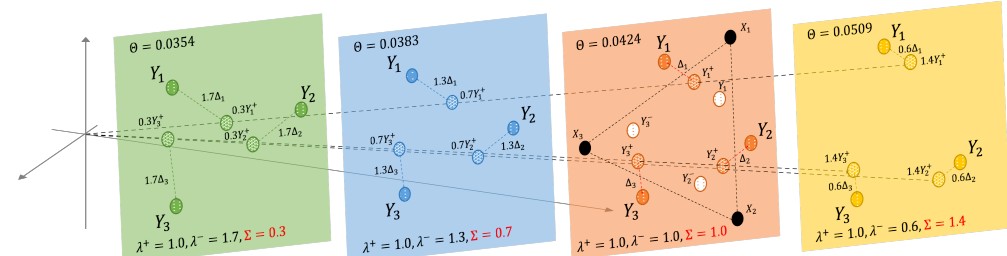

Figure 3: Different output space and representations according to $\lambda$ combinations. $\lambda^-$ varies while $\lambda^+$ is fixed to 1.

space determined by the processed representations from lower layers. This constraint enhances the influence of lower layers on the information processing of upper layers, thereby improving overall model efficacy.

Secondly, we examine the dynamic of output representations in relation to $\lambda$s. Utilizing the definitions established in Eqs.7 and 8, we can express $\mathbf{Y}$ in an alternative form as follows: $\mathbf{Y} = (1+\lambda^+)(\mathbf{P}^+\mathbf{X}_V) - \lambda^-(\mathbf{P}^-\mathbf{X}_V) = (1+\lambda^+)\mathbf{Y}^+ - \lambda^-\mathbf{Y}^-$. Here, $\mathbf{Y}^+$ represents the output of conventional attention mechanism, and both $\mathbf{Y}^+$ and $\mathbf{Y}^-$ are derived from convex combination of $\mathbf{X}_V$, as depicted in the third (orange) hyperplane of Fig.3. We can further reformulate this as follows: $\mathbf{Y} = \Sigma\mathbf{Y}^+ + \lambda^-(\mathbf{Y}^+ - \mathbf{Y}^-) = \Sigma\mathbf{Y}^+ + \lambda^-\Delta$. A crucial characteristic of $\Delta$ is that it represents a direct movement on the hyperplane from the original point $\mathbf{Y}^+$, and it is manipulated by $\lambda^-$, while the original point $\mathbf{Y}^+$ is manipulated by $\Sigma$. Consequently, when $\lambda^-$ is large to make $\Sigma$ less than 1, daGPAM scales-down the original point while amplifying the movement on the hyperplane, resulting in an increase in the relative diversity of output representations compared to their average norm. This dynamic is illustrated in Fig.3 with empirically computed cosine similarity, $\theta$, based on our toy experiment setting[3]. It shows that the cosine similarity decreases with only changing $\lambda^-$ to make a small $\Sigma$.

### 4.3 Theoretic Advantages of Dual-Attention GPAM

In this section, we discuss the theoretical advantages of daGPAM in addressing the issues of rank-collapse and gradient vanishing. First, to enhance our theoretical understanding of the rank-collapse problem, we derive the input/output relationship of the *residual*, Eq.4, based on daGPAM structure. The results of this derivation is presented in the following lemma.

**Lemma 4** (Dual-Attention GPAM *residual* Bound, Simplified). *For any single daGPAM self-attention layer with for a term $\gamma$ that depends on the attention entries, the composite norm of output residual is bounded by*

$$\|res(\mathbf{Y})\|_{1,\infty} \leq B^{org} + \frac{4\sqrt{2}\gamma\left(\left|\lambda^+\|\mathbf{W}_{QK}^+\|_1 - \lambda^-\|\mathbf{W}_{QK}^-\|_1\right|\right)\|\mathbf{W}_V\|_{1,\infty}}{\sqrt{d_{qk}}}\|res(\mathbf{X})\|_{1,\infty}^3, \quad (11)$$

*where $\mathbf{W}_{QK}^+ = \mathbf{W}_Q^+(\mathbf{W}_K^+)^\top$ and $\mathbf{W}_{QK}^- = (\mathbf{W}_Q^+\mathbf{W}_Q^-)(\mathbf{W}_K^+)^\top$. $B^{org}$ is the upper bound derived by Lemma 1. Because the second term is positive, this upper bound is always greater than the original.*

The complete lemma and its proof are provided in Appendix A.1.3. Based on this lemma, we argue that daGPAM can have greater or equal *residual* bound compared to the conventional attention mechanism, which suggests a higher likelihood that the average distance between output representations is either maintained or not reduced compared to the average distance of input representations.

Secondly, we derive the gradients of daGPAM. Due to the complexity of this derivation, throughout this derivation, we consider the simple case that approximates the negative unnormalized attention

---

[3]We implemented daGPAM, Eqs.7∼10, with $\mathbf{X}$ and the weight matrices which are randomly initialized by standard Gaussian distribution. Then, we computed the average cosine similarity of each output representations $\mathbf{Y}$ with each $\lambda$ combination over 100 times repeatedly.

score is simply the positive score with -1 multiplication, that is $\mathbf{A}^+ = -\mathbf{A}^-$ with approximating $\sigma$ to identity activation and $\mathbf{W}_Q^- \approx -\mathbf{I}$ where $\mathbf{I}$ is identity matrix with size $d_{qk}$. . The result of this derivation is as follows (we simply denote $\mathbf{A}^+$ as $\mathbf{A}$ for comparison between the original lemma 2):

**Lemma 5** (Dual-Attention GPAM Gradients). *The gradient that the input unnormalized attention score,* $\mathbf{A}$*, receives through the normalized attention score,* $\mathbf{P}^G$ *(without* $1/\sqrt{d_{qk}}$*) is derived as follows:*

$$\frac{\partial \mathbf{P}_{ij}^G}{\partial \mathbf{A}_{ij}} = g_j^{org} + \lambda^+ \mathbf{P}_{ij}^+(1 - \mathbf{P}_{ij}^+) + \lambda^- \mathbf{P}_{ij}^-(1 - \mathbf{P}_{ij}^-), \tag{12}$$

$$\frac{\partial \mathbf{P}_{ik,k\neq j}^G}{\partial \mathbf{A}_{ij}} = g_k^{org} + \lambda^+(-\mathbf{P}_{ik}^+ \mathbf{P}_{ij}^+) + \lambda^-(-\mathbf{P}_{ik}^- \mathbf{P}_{ij}^-), \tag{13}$$

*where* $g_j^{org}$ *and* $g_k^{org}$ *are the derived gradient of the conventional attention mechanism, Eq.6, respectively.*

The proof of this lemma is provided in Appendix A.1.4. Because the last two additional terms have the same sign of the original, daGPAM always flow greater gradients than the conventional attention mechanism. Therefore, daGPAM can mitigate the rank-collapse and gradient vanishing problems together.

## 5 EMPIRICAL VALIDATIONS

To evaluate the effectiveness of daGPAM in addressing the rank-collapse problem relative to the conventional attention mechanism, we conducted rank-collapse analyses at initialization (faithfulness test) (Poole et al., 2016; Schoenholz et al., 2016; Yang & Schoenholz, 2017; Hayou et al., 2019; Noci et al., 2024) and after the training phase. To assess the impact on the gradient vanishing problem, we monitored the gradient norm history of the query weights, specifically $\|\frac{\partial L}{\partial \mathbf{W}_Q}\|_2$, throughout the training process. Finally, to argue the importance of preserving generalized probability conditions, as discussed in Section 2.3, we compare daGPAM with other several alternative attention mechanisms in preliminary experiments.

All empirical analyses were conducted within the framework of our preliminary experimental settings, specifically utilizing the Penn Treebank dataset (PTB, 1M total number of tokens (Marcus et al., 1993)), for a world-level LM task using a 15-layered decoder-only Transformer model. We employed open-source resource[4] for data-related processes, including preprocessing and tokenization. Detailed configurations regarding the model architecture and optimization processes are presented in Table 4 in Appendix A.2. We replaced only the conventional attention layers, self-attention layer, with daGPAM. We explored various configurations, including different combinations of constant $\lambda$s and trainable $\lambda$ methods.

### 5.1 RANK-COLLAPSE ANALYSES

For the two rank-collapse analyses, we measured the amount of diversity based on the output representations of self-attention layer (output of multi-head attention layer) for all layers (15, in this experiment). We used two different metrics to measure the diversity.

- Average relative norm of '*residual*' (Dong et al., 2021) : $res(\mathbf{Y}) = \frac{1}{T}\sum_{i=1}^{T} \frac{\|\mathbf{Y}_i - \bar{\mathbf{y}}\|_2}{\|\mathbf{Y}_i\|_2}$.

- Average *cosine similarity* (Noci et al., 2022): $cos(\mathbf{Y}) = \frac{1}{T^2}\sum_{i=1}^{T}\sum_{j=1}^{T} \frac{\mathbf{Y}_i \cdot \mathbf{Y}_j}{\|\mathbf{Y}_i\|_2\|\mathbf{Y}_i\|_2}$.

Higher or lower values of $res(\mathbf{Y})$ or $cos(\mathbf{Y})$, respectively, mean less collapsed representations $\mathbf{Y}$.

The left two columns of Fig.4 present the results of the two rank-collapse analyses. In the analysis at initialization, as known, the rank-collapse phenomenon becomes more pronounced in the upper layers. However, except for the case with $(\lambda^+ = 1.0, \lambda^- = 0.5)$, daGPAM models exhibited a less intensive tendency toward rank-collapse. Notably, this mitigation becomes more effective when $\Sigma$ is

---

[4]https://github.com/kimiyoung/transformer-xl/

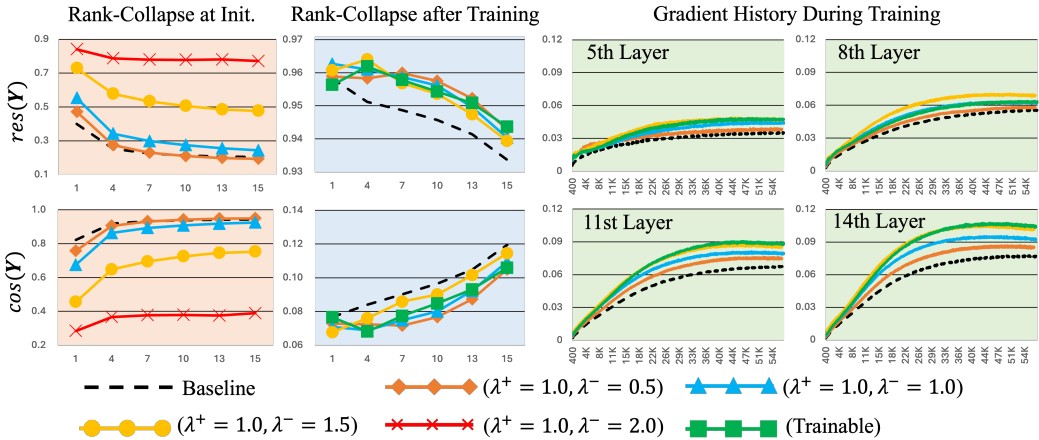

Figure 4: The results of rank-collapse analyses (left two graphs) and gradient histories during training (right two graphs). Horizontal axis of rank-collapse analyses indicate layer index, while those of gradient histories indicate training iterations.

Table 1: PTB experiment results of various attention mechanisms. Symbol '*' means that the initial numbers are trainable. 'X' means the model does not follow that condition, that is unbounded infinite range or non-fixed total sum of normalized attention scores.

| Model | # Param. | Finite Range of $\mathbf{P}$ | Fixed Sum $\Sigma$ | PTB (PPL) |
|---|---|---|---|---|
| Baseline Transformer | 29.2M | [0, 1] | 1 | 108.26 |
| NON (Richter & Wattenhofer, 2022) | 29.2M | X | X | 168.97 |
| NAP (Richter & Wattenhofer, 2022) | 29.2M | X | 0* | 258.85 |
| CoDA (Tay et al., 2019) | 29.2M | [-1, 1] | X | 133.11 |
| Sin-Softmax (Wang et al., 2021) | 29.2M | [0, 1] | 1 | 123.04 |
| ValueSkipInit (He et al., 2023) | 29.2M | [0, 2]* | 2* | 108.40 |
| Shaped (Noci et al., 2024) | 29.2M | $[-\frac{1}{T}, 2-\frac{1}{T}]$ | 1 | 109.06 |
| daGPAM ($\lambda^+ = 1.0$, $\lambda^- = 1.0$) | 29.6M | [-1, 2] | 1 | 109.01 |
| daGPAM (Trainable $\lambda$s) | 29.6M | [-1, 2]* | 1* | **106.38** |

smaller, such as in the case of $(\lambda^+ = 1.0, \lambda^- = 2.0)$, where $\Sigma = 0$ showed the greatest reduction in rank-collapse. In the analysis after training phase, all daGPAM models demonstrated a less intensive tendency for rank-reduction compared to the baseline. These empirical findings are consistent with Lemma 4 and the discussions in Section 4.2. Additionally, we provide further analyses addressing other aspects beyond the scope of the above experiments, including the effect of varying $\lambda^+$, a rank-collapse analysis based on the output of the multi-layer perceptron (MLP) layer (i.e., the final output of the Transformer layer), and an examination of the actual influence of the attention layer compared to the shortcut branch. These additional results can be found in Appendix A.3.

## 5.2 GRADIENT HISTORY DURING TRAINING

In addition to addressing the rank-collapse issue, we tracked the gradient norm history of the query weight matrix to verify our claim regarding the gradient vanishing problem (Lemma 5). The right two columns of Fig.4 illustrate the gradient norms of query weights across layers 5, 8, 11, and 14. In each case, the query weights in daGPAM model exhibited larger gradients compared to the baseline, supporting the predictions in Lemma 5. Furthermore, the observation that higher layers receive stronger gradients might align with Lemma 3. As described in the lemma, significant rank-collapse leads to the maximum gradient flow through the softmax operation. Combining with the analysis result of higher layers' more collapsed output representations (Section 5.1), we guess the lemma could explain why upper layers receive greater gradients than lower layers.

Table 2: Experiments on wikitext103 and Enwiki8 LM tasks. 'daGPAM (Const)' and 'daGPAM (Train)' mean daGPAM models with pre-defined constant $\lambda$s and trainable $\lambda$s, respectively.

| Model | Wikitext103 | | | Enwiki8 | |
|---|---|---|---|---|---|
| | 8L | 16L | 24L | 6L | 12L |
| TransformerXL | 25.96 | 23.58 | 22.90 | 1.1570 | 1.0544 |
| daGPAM (Const) | 25.39 | **23.09** | **22.52** | 1.1534 | **1.0473** |
| daGPAM (Train) | **25.33** | 23.20 | 22.57 | **1.1532** | 1.0528 |

## 5.3 COMPARISON BETWEEN OTHER ALTERNATIVE ATTENTION MECHANISMS

In this section, we compare the performance of daGPAM with other alternative attention mechanisms discussed in Section 2.3. We specifically focus on analyzing each mechanism's adherence to the generalized probability conditions and its corresponding performance, in order to highlight the significance of adhering these conditions. For a fair comparison, we re-implemented all alternative attention mechanisms and applied them to the Transformer baseline, replacing the conventional attention mechanism in the same manner as with daGPAM.

Table 1 presents the results of various attention mechanisms trained on the PTB LM task, evaluated using perplexity (PPL). Our daGPAM models resulted in approximately 1% increases in parameters. Each model is labeled to indicate whether it adheres to the generalized probability conditions, specifically finite range and a fixed total sum of normalized attention scores. The models which do not adhere to the conditions exhibit significant performance degradation. Conversely, with the exception of 'Sin-Softmax,' most models that adhered to both conditions performed similarly to the baseline Transformer. Although the 'ValueSkipInit' and 'Shaped' models did not degrade performance, they also failed to provide improvements, because these methods regularize the attention matrix to be similar to the identity matrix, thereby weakening the contextualization effect. In contrast, daGPAM enhances the attention mechanism itself, leading to improved performance in the 'daGPAM (Trainable $\lambda$s)' model.

## 6 PRACTICAL BENEFITS IN BENCHMARK EXPERIMENTS

In this section, we compare the performance of our models on several benchmark tasks. We conducted LM experiments on the Wikitext103 (word-level LM) and Enwiki8 (character-level LM) datasets. We followed the same data-related processes based on the same open-source of the PTB dataset used in our preliminary experiment. For the baseline model, we re-implemented TransformerXL (Dai, 2019), heavily following the public code[5]. This model is larger than the standard Transformer and is optimized for long-context sentences. We followed the original congifurations of model and optimization, except for modifications in the number of layers and the hyperparameters related to daGPAM approach. For further details on the basic configurations, refer to (Dai, 2019).

Second, we performed NMT experiments on the IWSLT14 English-German dataset (160K training pairs) and the WMT14 English-German dataset (3.9M training pairs) (Heo et al., 2024). The data preprocessing steps, including tokenization and subword byte-pair encoding, were carried out using the Fairseq toolkit (Ott et al., 2019). For the baseline models, we re-implemented PreLN (Xiong et al., 2020) and Admin (Liu et al., 2020) encoder-decoder Transformer architectures, which are widely used NMT baselines nowadays. The basic model configurations and optimization settings are provided in Table 4 of Appendix A.2. Throughout our experiments with daGPAM models (including models in LM task) using constant $\lambda$ values, we empirically determined the optimal combination of $\lambda$s, which are reported in Tables 5 and 6 in Appendix A.2.

### 6.1 LANGUAGE MODELING

For the evaluation of LM experiments, we utilized PPL for the Wikitext103 (word-level) task and bits per character (BPC) for the Enwiki8 (character-level) task. The experiments were conducted by varying the number of layers in the TransformerXL model. For the TransformerXL baseline models,

---

[5]https://github.com/kimiyoung/transformer-xl/

Table 3: Experiments on IWSLT14 and WMT14 NMT tasks. 'daGPAM (Const)' and 'daGPAM (Train)' were individually applied to the two architectures, respectively: PreLN (at the first row) and Admin (at the second row)

| Model | IWSLT14 | | WMT14 | |
|---|---|---|---|---|
| | En-to-De | De-to-En | En-to-De | De-to-En |
| PreLN | 28.54 | 33.90 | 26.40 | 31.26 |
| daGPAM (Const) | **29.25** | **34.43** | 26.73 | 31.43 |
| daGPAM (Train) | 29.11 | 34.11 | **27.19** | **31.45** |
| Admin | 28.32 | 33.48 | 26.27 | 30.61 |
| daGPAM (Const) | **28.60** | **33.62** | 26.76 | **31.20** |
| daGPAM (Train) | 28.43 | 33.23 | **26.79** | 31.10 |

the number of parameters are (130.6M / 151.1M / 171.6M) and (10.3M / 41.1M) for Wikitext103 (8L / 16L / 24L) and Enwiki8 (6L / 12L), respectively. In comparison, the number of parameters for daGPAM models are (130.7M / 151.3M / 172.0M) and (10.4M / 41.4M) for the tasks, respectively. On average, our approach increased only 0.43% of total parameters.

Table 2 demonstrates the results of LM experiments. It shows that daGPAM models consistently improve performance across different layered model architectures. Specifically, our best-performing models improve approximately 0.5 PPL on the Wikitext103 task and 0.0055 BPC on the Enwiki8 task in average. Additionally, we conducted an analysis of the impact of different $\lambda$ combinations within daGPAM model, with results detailed in Appendix A.3.4. Notably, we found that configurations where $\Sigma$ is close to 0.5 yielded the optimal performance in the Wikitext103 LM task. Also, the average $\Sigma$ calculated from the trained $\lambda$ values in the 'daGPAM (Train)' (8L) is 0.3948, with $\lambda^+ = 0.3303$ and $\lambda^- = 0.9355$.

## 6.2 Neural Machine Translation

For the evaluation of NMT experiments, we utilized case-sensitive SacreBLEU (Post, 2018). For the baseline models, the number of parameters are 64.67M and 153.84M for the IWSLT14 and WMT14 tasks, respectively. In comparison, the number of parameters for daGPAM models are 65.26M and 155.02M parameters, which is around 0.84% addition in average. Table 3 presents the results of the NMT experiments. Similar to the results of the LM experiments, daGPAM models consistently outperformed the baseline models. Specifically, our best-performing models demonstrated, in average, 0.42 BLEU point improvement for the IWSLT14 task and 0.52 BLEU points for the WMT14 task.

## 7 Conclusion

In this paper, we proposed a novel class of attention mechanism, generalized probabilistic attention mechanism (GPAM), which allows the negative attention score during the information processing. Also, we proposed one specific type of GPAM, dual-attention GPAM (daGPAM). While showing that the rank-collapse and gradient vanishing problems in the conventional attention mechanism is in trade-off relationship, we showed that daGPAM could mitigate both problems. Our empirical validations provide strong evidences supporting our theories and understanding of this approach. Additionally, our benchmark experiments demonstrate meaningful performance improvements with only a minimal increase in the number of parameters.

## 8 Future Works

Although we proposed only one structure of GPAM (daGPAM), GPAM does not have to be limited to the structure. Actually, the operation of daGPAM increases computational cost due to the added scaled dot-product attention process of negative part. In future, we aim to develop a more efficient structure than daGPAM while preserving the feature of GPAM. Also, beyond the standard Transformer architecture discussed here, GPAM can be applied to other architectures, such as graph neural networks and vision Transformer, as they are also known to experience the rank-collapse problem.

ACKNOWLEDGMENTS

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

# A APPENDIX

## A.1 PROOFS OF LEMMAS

### A.1.1 FULL DESCRIPTION OF LEMMA 1

In this section, we describe the complete version of the simplified lemma 1. For the proof of this lemma, see (Dong et al., 2021).

**Lemma 6** (Dong et al. (2021)). *For any single scaled dot-product self-attention layer that holds* $|\mathbf{E}_{ij} - \mathbf{E}_{ij'}| \leq 1.256$ *for any* $(i, j, j')$ *where* $\mathbf{E} = res(\mathbf{X})\frac{\mathbf{W}_{QK}}{\sqrt{d_{qk}}}res(\mathbf{X})^{\top}$, *and with* $\gamma$ *that satisfies*

$$\sqrt{\sum_{i=1}^{T} \max_{j,j'} |\mathbf{E}_{ij} - \mathbf{E}_{ij'}|} \leq \gamma \sqrt{\max_{j,j'} \sum_{i=1}^{T} |\mathbf{E}_{ij} - \mathbf{E}_{ij'}|}, \text{ the composite norm of residual of its}$$

*output is bounded by*

$$\|res(\mathbf{Y})\|_{1,\infty} \leq \frac{4\sqrt{2}\gamma\|\mathbf{W}_{QK}\|_1\|\mathbf{W}_V\|_{1,\infty}}{\sqrt{d_{qk}}}\|res(\mathbf{X})\|_{1,\infty}^3, \tag{14}$$

*where* $\mathbf{W}_{QK} = \mathbf{W}_Q\mathbf{W}_K^{\top}$. *In the region that holds* $4\sqrt{2}\gamma\|\mathbf{W}_{QK}\|_1\|\mathbf{W}_V\|_{1,\infty} < \sqrt{d_{qk}}$, *the output residual norm is diminished compared to the cubic rate of input residual norm.*

### A.1.2 PROOF OF LEMMA 3

In this section, we prove our proposed Lemma 3 which was introduced as follows:

**Lemma 7** (Maximum Total Norm of Gradients). *The total norm of gradients,* $G(\mathbf{P}_i)$, *is maximized when* $\mathbf{P}_i$ *is uniform distribution, that is* $\mathbf{P}_i = [\frac{1}{T}, \frac{1}{T}, \cdots, \frac{1}{T}]$ *which is the case of complete rank-collapse.*

*Proof.* Based on Eq.6, $G(\mathbf{P}_i)$ is derived as follows:

$$
\begin{aligned}
G(\mathbf{P}_i) &= \sum_{j=1}^{T}\sum_{k=1}^{T}\left|\frac{\partial \mathbf{P}_{ik}}{\partial \mathbf{A}_{ij}}\right| = \sum_{j=1}^{T}\left(\sum_{k=1,k\neq j}^{T}|-\mathbf{P}_{ik}\mathbf{P}_{ij}| + \mathbf{P}_{ij}(1-\mathbf{P}_{ij})\right), \\
&= \sum_{j=1}^{T}\left(\sum_{k=1}^{T}\mathbf{P}_{ik}\mathbf{P}_{ij} + \mathbf{P}_{ij}(1-2\mathbf{P}_{ij})\right), \\
&= \sum_{j=1}^{T}\mathbf{P}_{ij}\sum_{k=1}^{T}\mathbf{P}_{ik} + \sum_{j=1}^{T}\mathbf{P}_{ij} - 2\sum_{j=1}^{T}(\mathbf{P}_{ij})^2, \\
&= 2 - 2\sum_{j=1}^{T}(\mathbf{P}_{ij})^2.
\end{aligned}
\tag{15}
$$

Then, our goal is to find the input probability distribution, $\mathbf{P}_i$, that maximizes Eq.15, with constraint $\sum_{k=1}^{T}\mathbf{P}_{ik} = 1$. We use Lagrange's multiplier method to optimize this constrained maximization problem. With Lagrange coefficient, $\zeta$, we derive the new objective to maximize: $L(\mathbf{P}_i, \zeta) = 2 - 2\sum_{j=1}^{T}(\mathbf{P}_{ij})^2 - \zeta(\sum_{k=1}^{T}\mathbf{P}_{ik} - 1)$. The Jacobian of $L(\mathbf{P}_i, \zeta)$ with respect to $(\mathbf{P}_i, \zeta)$ is formulated as follows:

$$
\mathbb{J}_L = \begin{bmatrix} \frac{\partial L}{\partial \mathbf{P}_{i1}} & \cdots & \frac{\partial L}{\partial \mathbf{P}_{iT}} & \frac{\partial L}{\partial \zeta} \end{bmatrix} = \begin{bmatrix} (-4\mathbf{P}_{i1} - \zeta) & \cdots & (-4\mathbf{P}_{iT} - \zeta) & (-\sum_{k=1}^{T}\mathbf{P}_{ik} + 1) \end{bmatrix}.
$$

With setting the Jacobian to zero, the solutions are $\mathbf{P}_{ij} = -\frac{\zeta}{4}$ and $\zeta = -\frac{4}{T}$. Thus, an optimal value of $G(\mathbf{P}_i)$ is achieved when $\mathbf{P}_i^* = [\frac{1}{T}, \frac{1}{T}, \cdots, \frac{1}{T}]$.

To verify this point is the maximum, we compare $G(\mathbf{P}_i^*)$ with the total magnitude given slightly noised probabilities, $\mathbf{P}_i^{\epsilon_{x,y}}$ whose $x$-th probability is $\frac{1}{T} + \epsilon$, and $y$-th probability is $\frac{1}{T} - \epsilon$, where $x, y < T$ and $0 < \epsilon < \frac{1}{T}$. The two values are obtained as follows:

$$
\begin{aligned}
G(\mathbf{P}_i^*) &= 2 - 2\sum_{j=1}^{T}\left(\frac{1}{T}\right)^2 = 2 - 2\frac{1}{T}, \\
G(\mathbf{P}_i^{\epsilon_{x,y}}) &= 2 - 2\left(\sum_{j=1,j\neq x,y}^{T}\left(\frac{1}{T}\right)^2 + \left(\frac{1}{T}+\epsilon\right)^2 + \left(\frac{1}{T}-\epsilon\right)^2\right), \\
&= 2 - 2\left(\sum_{j=1}^{T}\left(\frac{1}{T}\right)^2 + 2\epsilon^2\right), \\
&= 2 - 2\frac{1}{T} - 4\epsilon^2 < G(\mathbf{P}_i^*).
\end{aligned}
$$

We can get the same result with any different combination of $x$ and $y$. Analogously, adding (and subtracting) multiple $\epsilon_i$ outputs lower than the optimal value. Therefore, $G(\mathbf{P}_i^*)$ is the maximum.

$\square$

### A.1.3 Full Description and Proof of Lemma 4

**Lemma 8** (Dual-Attention GPAM *residual* Bound, Completed). *Based on the constraints of Lemma 6 that are similarly applied to both positive/negative attention parts and jointly*

$$
\sqrt{\sum_{i=1}^{T}\max_{j,j'}|\mathbf{E}_{ij}^+ - \mathbf{E}_{ij'}^+|\max_{j,j'}|\mathbf{E}_{ij}^- - \mathbf{E}_{ij'}^-|} \leq \sqrt{2}\gamma\sqrt{\max_{j_1,j_1',j_2,j_2'}\sum_{i=1}^{T}|\mathbf{E}_{ij_1}^+ - \mathbf{E}_{ij_1'}^+||\mathbf{E}_{ij_2}^- - \mathbf{E}_{ij_2'}^-|},
$$

*the output residual's composite norm of any single daGPAM self-attention layer is bounded by*

$$\|res(\mathbf{Y})\|_{1,\infty} \leq \frac{4\sqrt{2}\gamma \left( \|\mathbf{W}_{QK}^+\|_1 + \left| \lambda^+ \|\mathbf{W}_{QK}^+\|_1 - \lambda^- \|\mathbf{W}_{QK}^-\|_1 \right| \right) \|\mathbf{W}_V\|_{1,\infty}}{\sqrt{d_{qk}}} \|res(\mathbf{X})\|_{1,\infty}^3,$$

$$= B^{org} + \frac{4\sqrt{2}\gamma \left( \left| \lambda^+ \|\mathbf{W}_{QK}^+\|_1 - \lambda^- \|\mathbf{W}_{QK}^-\|_1 \right| \right) \|\mathbf{W}_V\|_{1,\infty}}{\sqrt{d_{qk}}} \|res(\mathbf{X})\|_{1,\infty}^3, \quad (16)$$

*where* $\mathbf{W}_{QK}^+ = \mathbf{W}_Q^+ (\mathbf{W}_K^+)^\top$ *and* $\mathbf{W}_{QK}^- = (\mathbf{W}_Q^+ \mathbf{W}_Q^-)(\mathbf{W}_K^+)^\top$ *with assumption that the non-linear ReLU activation* $\sigma$ *is identity.* $B^{org}$ *is the upper bound derived by Lemma 6. Since the second term is positive, this upper bound is always greater than the original.*

*Proof.* In this proof, we follow the derivations in (Dong et al., 2021), except the triangle inequality for the Frobenius norm formulation in the middle.

By the technique described in Sec.A.4.2, the positive and negative unnormalized attention matrices, Eqs.9 and 10, are approximated as follows:

$$\mathbf{A}^+ \approx \frac{1}{\sqrt{d_{qk}}} \left( \mathbf{R}\mathbf{W}_{QK}^+\mathbf{R}^\top + \mathbf{1}\bar{\mathbf{x}}^\top \mathbf{W}_{QK}^+\mathbf{R}^\top \right),$$

$$\mathbf{A}^- \approx \frac{1}{\sqrt{d_{qk}}} \left( \mathbf{R}\mathbf{W}_{QK}^-\mathbf{R}^\top + \mathbf{1}\bar{\mathbf{x}}^\top \mathbf{W}_{QK}^-\mathbf{R}^\top \right),$$

where $\mathbf{W}_{QK}^+ = \mathbf{W}_Q^+ (\mathbf{W}_K^+)^\top$ and $\mathbf{W}_{QK}^- = (\mathbf{W}_Q^+ \mathbf{W}_Q^-)(\mathbf{W}_K^+)^\top$ with assumption that the non-linear ReLU activation $\sigma$ is identity, and $\mathbf{R} = res(\mathbf{X}) = \mathbf{X} - \mathbf{1}\bar{\mathbf{x}}^\top$. Then, the positive and negative normalized attention score matrices are formulated as follows:

$$\mathbf{P}^+ = softmax \left( \mathbf{E}^+ + \mathbf{1}(\mathbf{r}^+)^\top \right),$$

$$\mathbf{P}^- = softmax \left( \mathbf{E}^- + \mathbf{1}(\mathbf{r}^-)^\top \right),$$

where $\mathbf{E}^+ = \frac{1}{\sqrt{d_{qk}}}\mathbf{R}\mathbf{W}_{QK}^+\mathbf{R}^\top$, $\mathbf{E}^- = \frac{1}{\sqrt{d_{qk}}}\mathbf{R}\mathbf{W}_{QK}^-\mathbf{R}^\top$, $\mathbf{r}^+ = \frac{1}{\sqrt{d_{qk}}}\mathbf{R}(\mathbf{W}_{QK}^+)^\top\bar{\mathbf{x}}$, and $\mathbf{r}^- = \frac{1}{\sqrt{d_{qk}}}\mathbf{R}(\mathbf{W}_{QK}^-)^\top\bar{\mathbf{x}}$.

Following the technique described in Sec.A.4.3, each normalized attention matrix is lower and upper bounded as follows:

$$\left( \mathbf{I} - 2\mathbf{D}^+ \right)\mathbf{1}softmax(\mathbf{r}^+)^\top \leq \mathbf{P}^+ \leq \left( \mathbf{I} + 2\mathbf{D}^+ \right)\mathbf{1}softmax(\mathbf{r}^+)^\top,$$

$$\left( \mathbf{I} - 2\mathbf{D}^- \right)\mathbf{1}softmax(\mathbf{r}^-)^\top \leq \mathbf{P}^- \leq \left( \mathbf{I} + 2\mathbf{D}^- \right)\mathbf{1}softmax(\mathbf{r}^-)^\top,$$

where $\mathbf{D}$ is a diagonal matrix whose $i$-th diagonal element is $\mathbf{D}_{ii} = \max_{j,j'} |\mathbf{E}_{ij} - \mathbf{E}_{ij'}|$. Based on the design of daGPAM, Eq.8, $\mathbf{P}^G = (1 + \lambda^+)\mathbf{P}^+ - \lambda^-\mathbf{P}^-$, its lower and upper bound is formulated as follows:

$$(1 + \lambda^+)\left( \mathbf{I} - 2\mathbf{D}^+ \right)\mathbf{1}softmax(\mathbf{r}^+)^\top - \lambda^-\left( \mathbf{I} - 2\mathbf{D}^- \right)\mathbf{1}softmax(\mathbf{r}^-)^\top$$

$$\leq \mathbf{P}^G \leq$$

$$(1 + \lambda^+)\left( \mathbf{I} + 2\mathbf{D}^+ \right)\mathbf{1}softmax(\mathbf{r}^+)^\top - \lambda^-\left( \mathbf{I} + 2\mathbf{D}^- \right)\mathbf{1}softmax(\mathbf{r}^-)^\top.$$

Now, the output of the single daGPAM self-attention layer, $\mathbf{Y} = \mathbf{P}^G \mathbf{X}_V = \mathbf{P}^G \mathbf{X} \mathbf{W}_V$, is derived as follows:

$$
\begin{aligned}
\mathbf{Y} &= \mathbf{P}^G \mathbf{X} \mathbf{W}_V, \\
&= \mathbf{P}^G (\mathbf{1}\bar{\mathbf{x}}^\top + \mathbf{R}) \mathbf{W}_V, \\
&= \mathbf{P}^G \mathbf{1}\bar{\mathbf{x}}^\top \mathbf{W}_V + \mathbf{P}^G \mathbf{R} \mathbf{W}_V, \\
&= \mathbf{1}\bar{\mathbf{x}}^\top \mathbf{W}_V + \mathbf{P}^G \mathbf{R} \mathbf{W}_V, \\
&\leq \mathbf{1}\bar{\mathbf{x}}^\top \mathbf{W}_V + [(1 + \lambda^+)\left(\mathbf{I} + 2\mathbf{D}^+\right)\mathbf{1} softmax(\mathbf{r}^+)^\top \\
&\quad - \lambda^- \left(\mathbf{I} + 2\mathbf{D}^-\right)\mathbf{1} softmax(\mathbf{r}^-)^\top]\mathbf{R}\mathbf{W}_V, \\
&= \left(\mathbf{1}\bar{\mathbf{x}}^\top + (1 + \lambda^+)\mathbf{1} softmax(\mathbf{r}^+)^\top \mathbf{R} - \lambda^- \mathbf{1} softmax(\mathbf{r}^-)^\top \mathbf{R}\right) \mathbf{W}_V \\
&\quad + 2\left((1 + \lambda^+)\mathbf{D}^+\mathbf{1} softmax(\mathbf{r}^+)^\top - \lambda^- \mathbf{D}^- \mathbf{1} softmax(\mathbf{r}^-)^\top\right) \mathbf{R}\mathbf{W}_V, \\
\mathbf{Y} - \mathbf{1}(\mathbf{r}')^\top &\leq 2\left((1 + \lambda^+)\mathbf{D}^+\mathbf{1} softmax(\mathbf{r}^+)^\top - \lambda^- \mathbf{D}^- \mathbf{1} softmax(\mathbf{r}^-)^\top\right) \mathbf{R}\mathbf{W}_V,
\end{aligned}
$$

where $\mathbf{r}' = \mathbf{W}_V^\top \left(\bar{\mathbf{x}} + (1 + \lambda^+)\mathbf{R}^\top softmax(\mathbf{r}^+) - \lambda^- \mathbf{R}^\top softmax(\mathbf{r}^-)\right)$. Analogously, the lower bound of $(\mathbf{Y} - \mathbf{1}(\mathbf{r}')^\top)$ is given by

$$
\mathbf{Y} - \mathbf{1}(\mathbf{r}')^\top \geq -2\left((1 + \lambda^+)\mathbf{D}^+\mathbf{1} softmax(\mathbf{r}^+)^\top - \lambda^- \mathbf{D}^- \mathbf{1} softmax(\mathbf{r}^-)^\top\right) \mathbf{R}\mathbf{W}_V.
$$

Following the definition of (Dong et al., 2021), we can see $(\mathbf{Y} - \mathbf{1}(\mathbf{r}')^\top)$ is an appropriate approximation of $res(\mathbf{Y}) = \mathbf{R}'$ with assuming $\mathbf{r}' \approx \arg\min_{\bar{\mathbf{x}}'} \|\mathbf{Y} - \mathbf{1}\bar{\mathbf{x}}'^\top\|$.

Based on the triangle inequality, the upper bound of the Frobenius norm of $\mathbf{R}'$, that is $\|\mathbf{R}'\|_F$, is obtained by

$$
\begin{aligned}
\|\mathbf{R}'\|_F &\leq 2\left\|\left((1 + \lambda^+)\mathbf{D}^+\mathbf{1} softmax(\mathbf{r}^+)^\top - \lambda^- \mathbf{D}^- \mathbf{1} softmax(\mathbf{r}^-)^\top\right)\right\|_F \|\mathbf{R}\|_F \|\mathbf{W}_V\|_F, \\
&= 2\sqrt{(1 + \lambda^+)^2 \|\mathbf{D}^+\mathbf{1} softmax(\mathbf{r}^+)^\top\|_F^2 + (\lambda^-)^2 \|\mathbf{D}^-\mathbf{1} softmax(\mathbf{r}^-)^\top\|_F^2} \\
&\quad \overline{+2\langle(1 + \lambda^+)\mathbf{D}^+\mathbf{1} softmax(\mathbf{r}^+)^\top, -\lambda^- \mathbf{D}^- \mathbf{1} softmax(\mathbf{r}^-)^\top\rangle_F} \sqrt{\|\mathbf{R}\|_F^2} \sqrt{\|\mathbf{W}_V\|_F^2} \\
&\leq 2\sqrt{(1 + \lambda^+)^2 \|\mathbf{D}^+\mathbf{1}\|_1 \|\mathbf{D}^+\mathbf{1}\|_\infty + (\lambda^-)^2 \|\mathbf{D}^-\mathbf{1}\|_1 \|\mathbf{D}^-\mathbf{1}\|_\infty} \\
&\quad \overline{-2(1 + \lambda^+)\lambda^- \langle\mathbf{D}^+\mathbf{1} softmax(\mathbf{r}^+)^\top, \mathbf{D}^- \mathbf{1} softmax(\mathbf{r}^-)^\top\rangle_F} \|\mathbf{R}\|_{1,\infty} \|\mathbf{W}_V\|_{1,\infty},
\end{aligned}
\tag{17}
$$

where $\|\cdot\|_{1,\infty} = \sqrt{\|\cdot\|_1 \|\cdot\|_\infty} \geq \sqrt{\|\cdot\|_F^2}$ by the Schatten norm inequality. The last inequality is obtained by the upper bounds described in Sec. A.4.4. $\langle\cdot,\cdot\rangle_F$ is the Frobenius inner product.

Based on the derived upper bound described in Sec.A.4.5, we derive following inequalities:

$$
\begin{aligned}
\|\mathbf{D}^+\mathbf{1}\|_1 \|\mathbf{D}^+\mathbf{1}\|_\infty &\leq 8\gamma^2 \|\mathbf{E}^+\|_1^2, \\
\|\mathbf{D}^-\mathbf{1}\|_1 \|\mathbf{D}^-\mathbf{1}\|_\infty &\leq 8\gamma^2 \|\mathbf{E}^-\|_1^2,
\end{aligned}
$$

where $\gamma$ jointly holds the two initial conditions $\sqrt{\sum_{i=1}^T \max_{j,j'} |\mathbf{E}_{ij}^+ - \mathbf{E}_{ij'}^+|} \leq$ $\gamma\sqrt{\max_{j,j'} \sum_{i=1}^T |\mathbf{E}_{ij}^+ - \mathbf{E}_{ij'}^+|}$ and $\sqrt{\sum_{i=1}^T \max_{j,j'} |\mathbf{E}_{ij}^- - \mathbf{E}_{ij'}^-|} \leq \gamma\sqrt{\max_{j,j'} \sum_{i=1}^T |\mathbf{E}_{ij}^- - \mathbf{E}_{ij'}^-|}$.

The Frobenius inner product term in the inequality is further derived as follows:

$$\langle \mathbf{D}^+ \mathbf{1} softmax(\mathbf{r}^+)^\top, \mathbf{D}^- \mathbf{1} softmax(\mathbf{r}^-)^\top \rangle_F = \sum_i^T \sum_j^T \mathbf{D}_{ii}^+ softmax(\mathbf{r}^+)_j \mathbf{D}_{ii}^- softmax(\mathbf{r}^-)_j,$$

$$= \left( \sum_i^T \mathbf{D}_{ii}^+ \mathbf{D}_{ii}^- \right) \left( \sum_j^T softmax(\mathbf{r}^+)_j softmax(\mathbf{r}^-)_j \right),$$

$$\leq \left( \sum_i^T \mathbf{D}_{ii}^+ \mathbf{D}_{ii}^- \right) \left( 1 \sum_j^T softmax(\mathbf{r}^-)_j \right),$$

$$= \sum_i^T \mathbf{D}_{ii}^+ \mathbf{D}_{ii}^-,$$

$$= \sum_i^T \max_{j,j'} |\mathbf{E}_{ij}^+ - \mathbf{E}_{ij'}^+| \max_{j,j'} |\mathbf{E}_{ij}^- - \mathbf{E}_{ij'}^-|,$$

$$\leq 2\gamma^2 \max_{j_1,j_1',j_2,j_2'} \sum_i^T |\mathbf{E}_{ij_1}^+ - \mathbf{E}_{ij_1'}^+||\mathbf{E}_{ij_2}^- - \mathbf{E}_{ij_2'}^-|,$$

$$\leq 8\gamma^2 \max_{j_1,j_2} \sum_i^T |\mathbf{E}_{ij_1}^+||\mathbf{E}_{ij_2}^-|,$$

$$\leq 8\gamma^2 \left( \max_j \sum_i^T |\mathbf{E}_{ij}^+| \right) \left( \max_j \sum_i^T |\mathbf{E}_{ij}^-| \right),$$

$$= 8\gamma^2 \|\mathbf{E}^+\|_1 \|\mathbf{E}^-\|_1.$$

The inequality from 5-th to 6-th line is based on the initial condition of this lemma. The techniques used for the latter inequalities are similar to the techniques used in Sec. A.4.5.

Incorporating above derivations to Eq.17, $\|\mathbf{R}'\|_F$ is obtained by

$$\|\mathbf{R}'\|_F \leq 4\sqrt{2}\gamma \left( \sqrt{(1+\lambda^+)^2 \|\mathbf{E}^+\|_1^2 - 2(1+\lambda^+)\lambda^- \|\mathbf{E}^+\|_1 \|\mathbf{E}^-\|_1 + (\lambda^-)^2 \|\mathbf{E}^-\|_1^2} \right) \|\mathbf{R}\|_{1,\infty} \|\mathbf{W}_V\|_{1,\infty},$$

$$= 4\sqrt{2}\gamma \left( \left| ((1+\lambda^+)\|\mathbf{E}^+\|_1 - \lambda^- \|\mathbf{E}^-\|_1) \right| \right) \|\mathbf{R}\|_{1,\infty} \|\mathbf{W}_V\|_{1,\infty}, \tag{18}$$

By substituting $\mathbf{E}^+$ and $\mathbf{E}^-$ with their original form, respectively, we achieve the result of Lemma as follows:

$$\|\mathbf{R}'\|_F \leq 4\sqrt{2}\gamma \left( \left| (1+\lambda^+)\|\frac{1}{\sqrt{d_{qk}}} \mathbf{R}\mathbf{W}_{QK}^+ \mathbf{R}^\top \|_1 - \lambda^- \|\frac{1}{\sqrt{d_{qk}}} \mathbf{R}\mathbf{W}_{QK}^- \mathbf{R}^\top \|_1 \right| \right) \|\mathbf{R}\|_{1,\infty} \|\mathbf{W}_V\|_{1,\infty},$$

$$\leq \frac{4\sqrt{2}\gamma}{\sqrt{d_{qk}}} \left( \left| (1+\lambda^+)\|\mathbf{R}\|_1 \|\mathbf{W}_{QK}^+\|_1 \|\mathbf{R}^\top\|_1 - \lambda^- \|\mathbf{R}\|_1 \|\mathbf{W}_{QK}^-\|_1 \|\mathbf{R}^\top\|_1 \right| \right) \|\mathbf{R}\|_{1,\infty} \|\mathbf{W}_V\|_{1,\infty},$$

$$= \frac{4\sqrt{2}\gamma}{\sqrt{d_{qk}}} \left( \left| (1+\lambda^+)\|\mathbf{R}\|_1 \|\mathbf{W}_{QK}^+\|_1 \|\mathbf{R}\|_\infty - \lambda^- \|\mathbf{R}\|_1 \|\mathbf{W}_{QK}^-\|_1 \|\mathbf{R}\|_\infty \right| \right) \|\mathbf{R}\|_{1,\infty} \|\mathbf{W}_V\|_{1,\infty},$$

$$= \frac{4\sqrt{2}\gamma}{\sqrt{d_{qk}}} \left( \left| (1+\lambda^+)\|\mathbf{W}_{QK}^+\|_1 - \lambda^- \|\mathbf{W}_{QK}^-\|_1 \right| \right) \|\mathbf{R}\|_{1,\infty}^3 \|\mathbf{W}_V\|_{1,\infty},$$

$$\leq \frac{4\sqrt{2}\gamma}{\sqrt{d_{qk}}} \left( \|\mathbf{W}_{QK}^+\|_1 + \left| \lambda^+ \|\mathbf{W}_{QK}^+\|_1 - \lambda^- \|\mathbf{W}_{QK}^-\|_1 \right| \right) \|\mathbf{R}\|_{1,\infty}^3 \|\mathbf{W}_V\|_{1,\infty}.$$

$\square$

### A.1.4 PROOF OF LEMMA 5

In this section, we prove our proposed Lemma 5 which was introduced as follows:

**Lemma 9** (Dual-Attention GPAM Gradients). *The gradient that the input unnormalized attention score, $\mathbf{A}$, receives through the normalized attention score, $\mathbf{P}^G$ (without $1/\sqrt{d_{qk}}$) is derived as follows:*

$$\frac{\partial \mathbf{P}_{ij}^G}{\partial \mathbf{A}_{ij}} = (1 + \lambda^+)\mathbf{P}_{ij}^+(1 - \mathbf{P}_{ij}^+) + \lambda^-\mathbf{P}_{ij}^-(1 - \mathbf{P}_{ij}^-),$$

$$= g_j^{org} + \lambda^+\mathbf{P}_{ij}^+(1 - \mathbf{P}_{ij}^+) + \lambda^-\mathbf{P}_{ij}^-(1 - \mathbf{P}_{ij}^-), \tag{19}$$

$$\frac{\partial \mathbf{P}_{ik,k\neq j}^G}{\partial \mathbf{A}_{ij}} = (1 + \lambda^+)(-\mathbf{P}_{ik}^+\mathbf{P}_{ij}^+) + \lambda^-(-\mathbf{P}_{ik}^-\mathbf{P}_{ij}^-),$$

$$= g_k^{org} + \lambda^+(-\mathbf{P}_{ik}^+\mathbf{P}_{ij}^+) + \lambda^-(-\mathbf{P}_{ik}^-\mathbf{P}_{ij}^-), \tag{20}$$

*where $g_j^{org}$ and $g_k^{org}$ are the derived gradient of the conventional attention mechanism, Eq.6, respectively.*

*Proof.* Basically, the final gradients are simply formulated as follows:

$$\frac{\partial \mathbf{P}_{ij}^G}{\partial \mathbf{A}_{ij}} = (1 + \lambda^+)\frac{\partial \mathbf{P}_{ij}^+}{\partial \mathbf{A}_{ij}} - \lambda^-\frac{\partial \mathbf{P}_{ij}^-}{\partial \mathbf{A}_{ij}},$$

$$\frac{\partial \mathbf{P}_{ik,k\neq j}^G}{\partial \mathbf{A}_{ij}} = (1 + \lambda^+)\frac{\partial \mathbf{P}_{ik,k\neq j}^+}{\partial \mathbf{A}_{ij}} - \lambda^-\frac{\partial \mathbf{P}_{ik,k\neq j}^-}{\partial \mathbf{A}_{ij}}.$$

We use the results of the derived gradients in Lemma 2 for the partial gradients of positive parts, $\frac{\partial \mathbf{P}_{ij}^+}{\partial \mathbf{A}_{ij}}$ and $\frac{\partial \mathbf{P}_{ik,k\neq j}^+}{\partial \mathbf{A}_{ij}}$. The partial gradients of negative parts are derived as follows ($\mathbf{W}$ is the same as $\mathbf{W}_Q^-$ that we simplified as negative identity matrix):

$$\frac{\partial \mathbf{P}_{ij}^-}{\partial \mathbf{A}_{ij}} = \mathbf{W}_{jj}\frac{e^{\mathbf{W}_{jj}\mathbf{A}_{ij}^+}}{\sum_{t=1}^n e^{\mathbf{W}_{tt}\mathbf{A}_{it}^+}} - \frac{e^{\mathbf{W}_{jj}\mathbf{A}_{ij}^+}}{(\sum_{t=1}^n e^{\mathbf{W}_{tt}\mathbf{A}_{it}^+})^2}\left(\sum_{z=1}^n \mathbf{W}_{zj}e^{\mathbf{W}_{zz}\mathbf{A}_{iz}^+}\right),$$

$$= \mathbf{W}_{jj}\mathbf{P}_{ij}^- - \mathbf{P}_{ij}^-\left(\sum_{z=1}^n \mathbf{W}_{zj}\mathbf{P}_{iz}^-\right),$$

$$= \mathbf{W}_{jj}\mathbf{P}_{ij}^-(1 - \mathbf{P}_{ij}^-),$$

$$= -\mathbf{P}_{ij}^-(1 - \mathbf{P}_{ij}^-),$$

$$\frac{\partial \mathbf{P}_{ik,k\neq j}^-}{\partial \mathbf{A}_{ij}} = \mathbf{W}_{kj}\frac{e^{\mathbf{W}_{kj}\mathbf{A}_{ij}^+}}{\sum_{t=1}^n e^{\mathbf{W}_{tt}\mathbf{A}_{it}^+}} - \frac{e^{\mathbf{W}_{kj}\mathbf{A}_{ij}^+}}{(\sum_{t=1}^n e^{\mathbf{W}_{tt}\mathbf{A}_{it}^+})^2}\left(\sum_{z=1}^n \mathbf{W}_{zj}e^{\mathbf{W}_{zz}\mathbf{A}_{iz}^+}\right),$$

$$= \mathbf{W}_{kj}\mathbf{P}_{ik}^- - \mathbf{P}_{ik}^-\left(\sum_{z=1}^n \mathbf{W}_{zj}\mathbf{P}_{iz}^-\right),$$

$$= \mathbf{W}_{jj}(-\mathbf{P}_{ik}^-\mathbf{P}_{ij}^-),$$

$$= -(-\mathbf{P}_{ik}^-\mathbf{P}_{ij}^-),$$

During derivations, we used the simplification that $\mathbf{W}_{ij} = 0$ if $i \neq j$ and $\mathbf{W}_{ii} = -1$. Finally, the lemma is proved by substituting $\frac{\partial \mathbf{P}_{ij}^+}{\partial \mathbf{A}_{ij}}$, $\frac{\partial \mathbf{P}_{ik,k\neq j}^+}{\partial \mathbf{A}_{ij}}$, $\frac{\partial \mathbf{P}_{ij}^-}{\partial \mathbf{A}_{ij}}$, and $\frac{\partial \mathbf{P}_{ik,k\neq j}^-}{\partial \mathbf{A}_{ij}}$, from the above formulations with the newly derived formulations.

$\square$

### A.2 EXPERIMENTAL SETTINGS

In this section, we describe the experimental settings of our experiments. Except the benchmark LM experiments (Section 6.1), we used the described model and optimization configurations in Table 4

Table 4: Model and optimizer configurations used for our experiments. We tried to use the same notations with (Vaswani, 2017), except the number of layers (# of Layers) and multi-head attention's heads (# of Heads) for clarity. 'RAdam' means rectified Adam (Liu et al., 2019). 'ISRS' indicates inverse square root learning rate schedule (Ott et al., 2019) and '# of Tokens' indicates the number of tokens in a mini-batch at every iteration.

| Config. | PTB | IWSLT14 | | WMT14 | |
|---|---|---|---|---|---|
| | Transformer | PreLN | Admin | PreLN | Admin |
| $d_{model}$ | 256 | 512 | 512 | 512 | 512 |
| $d_{ff}$ | 2100 | 1024 | 1024 | 2048 | 2048 |
| $d_{qk}$ | 64 | 64 | 64 | 64 | 64 |
| $P_{drop}$ | 0.3 | 0.3 | 0.3 | 0.1 | 0.1 |
| $\epsilon_{ls}$ | 0.1 | 0.1 | 0.1 | 0.1 | 0.1 |
| # of Layers | 15 | 6 | 6 | 6 | 6 |
| # of Head | 4 | 4 | 4 | 8 | 8 |
| Optimizer | RAdam | RAdam | RAdam | RAdam | RAdam |
| Learning Rate | 0.00025 | 0.0005 | 0.0005 | 0.001 | 0.001 |
| Scheduler | ISRS | None | ISRS | None | ISRS |
| # of Tokens | 4K | 4K | 4K | 25K | 25K |
| Patience | 50 | 50 | 50 | 50 | 50 |

Table 5: Empirically found optimal $\lambda$ combination for our daGPAM model with constant $\lambda$s in LM experiments.

| Model | Wikitext103 | | | Enwiki8 | |
|---|---|---|---|---|---|
| | 8L | 16L | 24L | 6L | 12L |
| $(\lambda^+, \lambda^-)$ | (1.0, 1.5) | (1.5, 2.0) | (1.0, 1.0) | (1.0, 1.5) | (1.0, 1.0) |

for preliminary experiments (Section 5) and the benchmark NMT experiments (Section 6.2). For the experiments of benchmark LM experiments (Section 6.1), we followed the same experiment settings as the original work (Dai, 2019), except the different number of layers and other things related to our proposed daGPAM. For the settings of the optimal $\lambda$ combinations that we used for daGPAM (constant $\lambda$ setting) models, we demonstrated them in Tables 5 and 6.

During training of all experiments, we saved the best checkpoint based on validations. Especially, for NMT experiments, IWSLT14 and WMT14, we used checkpoint ensemble technique (Ott et al., 2019) with 10 latest checkpoints at each validation. Except the benchmark LM experiments, we early stopped the training whenever the model does not over the previous best performance 'Patience' times. For the experiments of benchmark LM experiments, we ran experiments with the original work's default training iteration settings.

We utilized single GTX1080Ti GPU for all of the preliminary experiments (Section 5). The PTB LM experiments took 12 hours in average. For all of our benchmark experiments (Section 6), we utilized single RTX3090 GPU. The TransformerXL-based LM experiments took averagely 20 and 240 hours for Wikitext-103 and Enwiki8 tasks, respectively. The NMT experiments took 75 and 120 hours in average for IWSLT14 and WMT14 tasks, respectively.

Table 6: Empirically found optimal $\lambda$ combination for our daGPAM model with constant $\lambda$s in NMT experiments.

| Model | IWSLT14 En-to-De | | IWSLT14 De-to-En | | WMT14 En-to-De | | WMT14 De-to-En | |
|---|---|---|---|---|---|---|---|---|
| | PreLN | Admin | PreLN | Admin | PreLN | Admin | PreLN | Admin |
| $(\lambda^+, \lambda^-)$ | (1.5, 1.5) | (2.0, 1.5) | (1.0, 1.5) | (2.0, 1.5) | (1.0, 1.0) | (1.0, 1.0) | (1.5, 1.5) | (1.0, 1.0) |

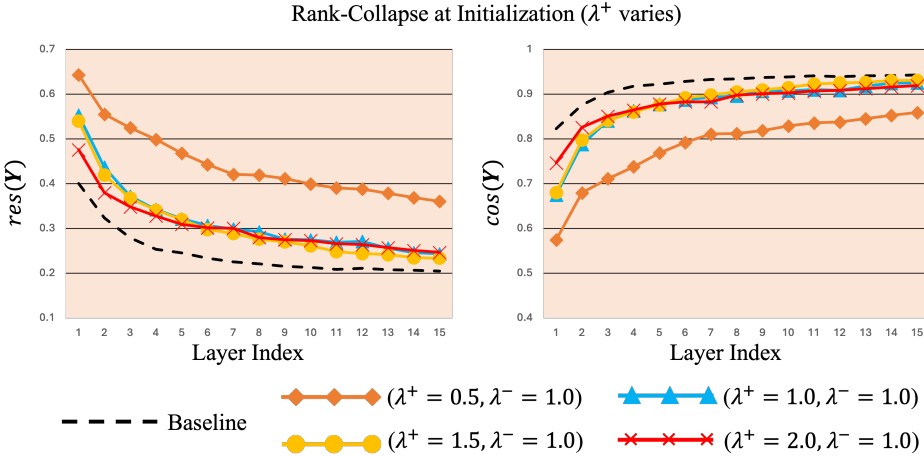

Figure 5: Results of faithfulness test (rank-collapse analysis at initialization) varying $\lambda^+$ with fixing $\lambda^-$ to 1.

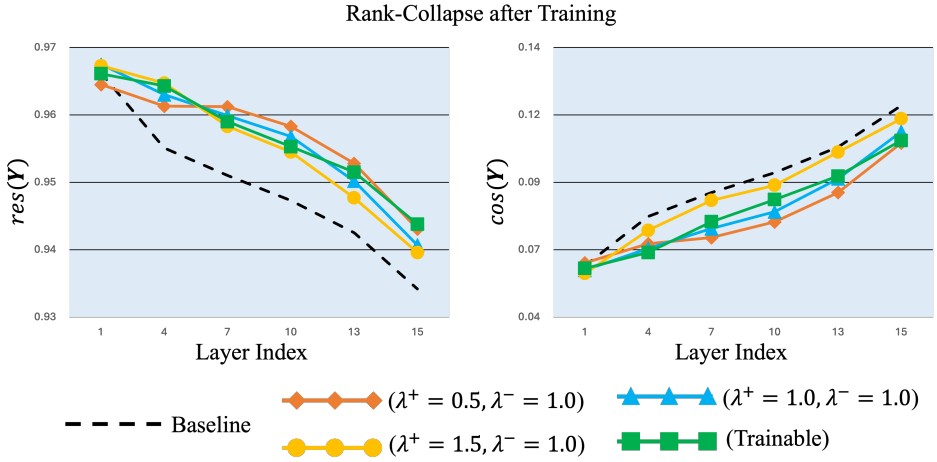

Figure 6: Results of rank-collapse analysis after training with MLP layer's output representations.

### A.3 Additional Experiment Results

#### A.3.1 Faithfulness Test with Varying $\lambda^+$

In addition to the rank-collapse analysis at initialization that is conducted with varying $\lambda^-$ while fixing $\lambda^+$ to 1 (Fig.4 in Section 5.1), we conducted the same analysis with varying $\lambda^+$ while fixing $\lambda^-$ to 1. As demonstrated in Fig.5, daGPAM models achieved less intensive rank-reduciton tendency than the baseline. Similar to the phenomenon explained in Section 5.1, daGPAM model that has $\Sigma$ smaller than 1 shows much less intensive tendency. For example, the $(\lambda^+ = 0.5, \lambda^- = 1.0)$ configuration shows the least intensive tendency, and this tendency is quite similar to the tendency of $(\lambda^+ = 1.0, \lambda^- = 1.5)$ configuration in Fig.4.

#### A.3.2 Rank-Collapse Analysis of MLP Layer's Output Representations

In addition to the rank-collapse analysis after training based on attention layer's output representations (Fig.4 in Section 5.1), we conducted the same analysis based on the MLP layer's output representations which are the final outputs of each Transformer block. As demonstrated in Fig.6, daGPAM models show mitigated rank-collapse phenomenon compared to the baseline, similar to the resulting phenomenon of attention layers' outputs.

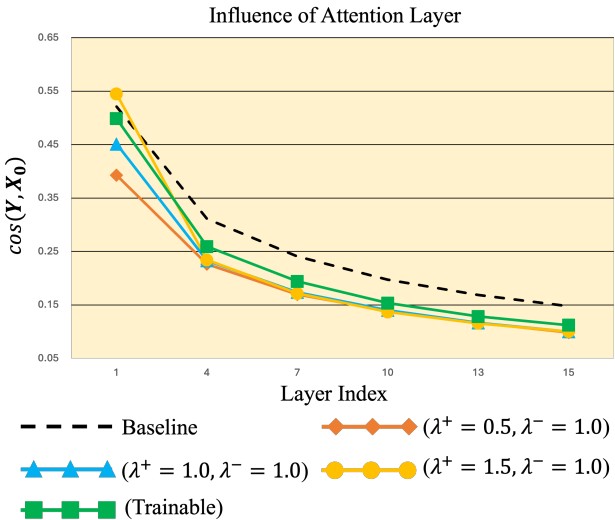

Figure 7: Cosine similarity results between the representations produced by the attention layer and the subsequent residual connection, relative to the initial representations. This cosine similarity serves as an indicator of the impact of the attention layer in comparison to the shortcut branch within the residual connection.

| $\lambda^-$ \ $\lambda^+$ | 0.5 | 1.0 | 1.5 | 2.0 | 2.5 | 3.0 |
|---|---|---|---|---|---|---|
| 0.5 | 25.82 | 25.63 | 25.62 | 25.51 | 25.49 | 25.41 |
| 1.0 | 25.70 | 25.53 | 25.62 | 25.61 | 25.43 | 25.46 |
| 1.5 | 25.75 | 25.39 | 25.59 | 25.60 | 25.43 | 25.40 |
| 2.0 | 25.60 | 25.75 | 25.37 | 25.31 | 25.38 | 25.39 |
| 2.5 | 25.58 | 25.41 | 25.85 | 25.37 | 25.31 | 25.47 |
| 3.0 | 25.79 | 25.82 | 25.56 | 25.72 | 25.32 | 25.46 |

Figure 8: PPL results of daGPAM models (based on 8-layered TransformerXL) with various constant $\lambda$s were trained on Wikitext103 LM task. The darker the color is, the better PPL is. The baseline model's PPL result is 25.96.

### A.3.3 INFLUENCE OF ATTENTION LAYERS

To assure that our proposed model develops the attention layer rather than enhancing the shortcut branch like some previous works (Noci et al., 2022), we measured the cosine similarity between residual connection's output representations and the initial representations (word embedding vectors). This cosine similarity means the influence of attention layer within the residual connection. As demonstrated in Fig.7, we found that daGPAM models usually achieve less cosine similarity which means the attention layers output representations are more diverse compared to the baseline's. Based on this analysis, we believe that daGPAM actually affects the attention layer rather than enhancing shortcut branch.

### A.3.4 WIKITEXT103 LM EXPERIMENTS WITH VARIOUS $\lambda$S

To provide profound understanding on the effects of $\lambda$s in practice, we conducted more Wikitext103 LM experiments of daGPAM models based on 8-layered TransformerXL architecture with varying each $\lambda$ from 0.5 to 3.0 with 0.5 interval. Note that the total sum of normalized attention score is $\Sigma = (1+\lambda^+-\lambda-)$. We discovered that, at each value of $\lambda^+$, increasing $\lambda^-$ improves performance until it makes $\Sigma$ lower than -2.0. Usually, the best performance was achieved when $\Sigma$ is 0.5. Interestingly, we found that, when $\Sigma$ is 0.0, performance instantly drops. We understand the 0.0 $\Sigma$ can overly

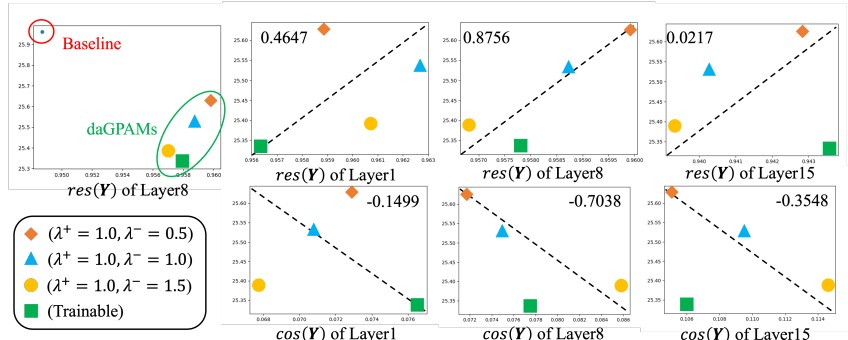

Figure 9: Correlation graphs between (*residual*, cosine similarity) in Fig.4 and PPL performances on Wikitext103 in Fig.8. The leftmost graph includes baseline model while the other 6 graphs include only daGPAM models. The four decimal place numbers displayed on each graph represent correlations.

amplify the direct movement on the hyperplane, $\Delta$, while eliminate the effect original point $\mathbf{Y}^+$ as we explained in Section 4.2. This may cause too diverse output representations. Empirically, we found that the cases when $\Sigma$ is close to 0.5, usually be the optimal setting for daGPAM models.

### A.3.5 Correlation between Rank-Collapse and Performance

To obtain profound understanding of the effect of mitigating rank-collapse problem in practice, we analyzed the correlation between the measured amount of rank-collapse (the second column of Fig.4) and the downstream performances of Wikitext103 results (various $\lambda^-$s with fixed $\lambda^+ = 1.0$ in Fig.8 and the result of 'daGPAM(Train)' in Table2). We assumed that the tendency for rank-collapse dependent on the setting of $\lambda$s as shown in Fig.4 would similarly appear in the Wikitext103 experiments. We note that daGPAM models mitigate not only rank-collapse, but also gradient vanishing, so we excluded the sample of baseline model and analyzed the correlations of daGPAM models to focus pure relationship of mitigating rank-collapse problem and downstream performance. The 6 graphs on the right side of Fig.9 show the computed correlations and graphs. We employed the measured rank-collapse amounts of lower layer 1, middle layer 8, and higher layer 15. We found that most rank-collapse amount measurements are highly correlated with downstream performance. While the *residual* of layer 15 shows almost no correlation, this is primarily due to the trainable $\lambda$ setting, which differs from other constant $\lambda$ settings.

## A.4 Technical Lemmas from (Dong et al., 2021)

### A.4.1 Properties of Matrix Norm

In this section, we describe the formulation and properties of matrix norm which are frequently used in our proof derivations. We note that L-1 and L-$\infty$ norms of a $T \times d$ matrix, $\mathbf{M}$, which are formulated as follow:

$$\|\mathbf{M}\|_1 = \max_{1 \leq i \leq d} \left( \sum_{j=1}^{T} |\mathbf{M}_{ji}| \right),$$

$$\|\mathbf{M}\|_\infty = \max_{1 \leq i \leq T} \left( \sum_{j=1}^{d} |\mathbf{M}_{ij}| \right).$$

Noticeably, $\|\mathbf{M}\|_1 = \|\mathbf{M}^\top\|_\infty$. If there is a matrix $\mathbf{N}$ which is multiplicative with $\mathbf{M}$, then $\|\mathbf{MN}\| \leq \|\mathbf{M}\|\|\mathbf{N}\|$. This property holds for L-1 and L-$\infty$ norms.

### A.4.2 Simplification of Unnormalized Attention Matrix

Given the definition of the unnormalized attention matrix $\mathbf{A}$, Eq.3, $\mathbf{A}$ is related to the *residual*, Eq.4, as follows:

$$\mathbf{A} = \frac{1}{\sqrt{d_{qk}}}(\mathbf{R} + \mathbf{1}\bar{\mathbf{x}}^\top)\mathbf{W}_{QK}(\mathbf{R} + \mathbf{1}\bar{\mathbf{x}}^\top)^\top$$

$$= \frac{1}{\sqrt{d_{qk}}}\left(\mathbf{R}\mathbf{W}_{QK}\mathbf{R}^\top + \mathbf{R}\mathbf{W}_{QK}\bar{\mathbf{x}}\mathbf{1}^\top + \mathbf{1}\bar{\mathbf{x}}^\top\mathbf{W}_{QK}\mathbf{R}^\top + \mathbf{1}\bar{\mathbf{x}}^\top\mathbf{W}_{QK}\bar{\mathbf{x}}\mathbf{1}^\top\right),$$

where $\mathbf{W}_{QK} = \mathbf{W}_Q\mathbf{W}_K^\top$ and $\mathbf{R} = res(\mathbf{X}) = \mathbf{X} - \mathbf{1}\bar{\mathbf{x}}^\top$. By following the shift invariance property of the softmax function (Cordonnier et al., 2020) which means a constant term added to every element of a row is negligible, we approximate $\mathbf{A}$ as follows:

$$\widetilde{\mathbf{A}} \approx \frac{1}{\sqrt{d_{qk}}}\left(\mathbf{R}\mathbf{W}_{QK}\mathbf{R}^\top + \mathbf{1}\bar{\mathbf{x}}^\top\mathbf{W}_{QK}\mathbf{R}^\top\right).$$

Note that every eliminated (by the approximation) terms has the form, $\mathbf{c}\mathbf{1}^\top$, so that we can express $\mathbf{A} = \widetilde{\mathbf{A}} + \mathbf{c}'\mathbf{1}^\top$ where the vector, $\mathbf{c}'$, is the summation of every eliminated terms without the common factor, $\mathbf{1}^\top$. Then, the output of the softmax function with $\mathbf{A}$ as input is $softmax(\mathbf{A}) = softmax(\widetilde{\mathbf{A}} + \mathbf{c}'\mathbf{1}^\top) = softmax(\widetilde{\mathbf{A}})$. Therefore, we can safely use the approximated unnormalized attention matrix instead of the original during the attention mechanism.

### A.4.3 Bounds of Normalized Attention Matrix

With incorporating the approximation of the unnormalized attention matrix to the definition of the normalized attention matrix, Eq.2 (without considerations of positive and negative signs), we can formulate the normalized attention matrix as follows:

$$\mathbf{P} = softmax(\frac{1}{\sqrt{d_{qk}}}\mathbf{A}) = softmax\left(\mathbf{1}\mathbf{r}^\top + \mathbf{E}\right),$$

where $\mathbf{E} = \frac{1}{\sqrt{d_{qk}}}\mathbf{R}\mathbf{W}_{QK}\mathbf{R}^\top$ and $\mathbf{r} = \frac{1}{\sqrt{d_{qk}}}\mathbf{R}(\mathbf{W}_{QK})^\top\bar{\mathbf{x}}$.

The $(i, j)$-th element of $\mathbf{P}$ is derived as follow:

$$\mathbf{P}_{ij} = \frac{\exp\left((\mathbf{1}\mathbf{r}^\top)_{ij} + \mathbf{E}_{ij}\right)}{\sum_{t=1}^T \exp\left((\mathbf{1}\mathbf{r}^\top)_{it} + \mathbf{E}_{it}\right)} = \frac{\exp\left((\mathbf{1}\mathbf{r}^\top)_{ij}\right)\exp\left(\mathbf{E}_{ij}\right)}{\sum_{t=1}^T \exp\left((\mathbf{1}\mathbf{r}^\top)_{it}\right)\exp\left(\mathbf{E}_{it}\right)}.$$

This value is upper bounded when we replace $\exp(\mathbf{E}_{it})$ in the denominator with $\min_{j'}\exp(\mathbf{E}_{ij'})$. Likewise, it is lower bounded when we replace the same term with $\max_{j'}\exp(\mathbf{E}_{ij'})$. Based on these ideas, we can bound the $(i, j)$-th element of $\mathbf{P}$ as follows:

$$\frac{\exp(\mathbf{E}_{ij})}{\max_{j'}\exp(\mathbf{E}_{ij'})}\widetilde{\mathbf{P}}_{ij} \leq \mathbf{P}_{ij} \leq \frac{\exp(\mathbf{E}_{ij})}{\min_{j'}\exp(\mathbf{E}_{ij'})}\widetilde{\mathbf{P}}_{ij},$$

$$\left(\min_{j'}\exp(\mathbf{E}_{ij} - \mathbf{E}_{ij'})\right)\widetilde{\mathbf{P}}_{ij} \leq \mathbf{P}_{ij} \leq \left(\max_{j'}\exp(\mathbf{E}_{ij} - \mathbf{E}_{ij'})\right)\widetilde{\mathbf{P}}_{ij},$$

where $\widetilde{\mathbf{P}}_{ij} = \frac{\exp((\mathbf{1}\mathbf{r}^\top)_{ij})}{\sum_{t=1}^T \exp((\mathbf{1}\mathbf{r}^\top)_{it})} = softmax(\mathbf{1}\mathbf{r}^\top)_{ij} = (\mathbf{1}softmax(\mathbf{r})^\top)_{ij}$. Given the fact that $\exp(x)$ is approximated by $1 + x + \frac{1}{2!}x^2 + \frac{1}{3!}x^3 + \dots$ with Taylor series near to $x = 0$ and it is upper bounded $\exp(x) \leq 1 + 2x$ where the condition $|x| \leq 1.256$ holds, we know that $\max_{j'}\exp(\mathbf{E}_{ij} - \mathbf{E}_{ij'})$ is upper bounded by $1 + 2\max_{j'}(\mathbf{E}_{ij} - \mathbf{E}_{ij'})$ with the condition that $|\mathbf{E}_{ij} - \mathbf{E}_{ij'}| \leq 1.256$. Analogously, $\min_{j'}\exp(\mathbf{E}_{ij} - \mathbf{E}_{ij'})$ is lower bounded by $1 - 2\min_{j'}(\mathbf{E}_{ij} - \mathbf{E}_{ij'})$. Therefore, the lower

and upper bounds of $(i, j)$-th element of $\mathbf{P}$ are derived as follows:

$$\left(1 - 2\min_{j'}(\mathbf{E}_{ij} - \mathbf{E}_{ij'})\right)\widetilde{\mathbf{P}}_{ij} \leq \mathbf{P}_{ij} \leq \left(1 + 2\max_{j'}(\mathbf{E}_{ij} - \mathbf{E}_{ij'})\right)\widetilde{\mathbf{P}}_{ij},$$

$$\left(1 + 2\max_{j'}(\mathbf{E}_{ij'} - \mathbf{E}_{ij})\right)\widetilde{\mathbf{P}}_{ij} \leq \mathbf{P}_{ij} \leq \left(1 + 2\max_{j'}(\mathbf{E}_{ij} - \mathbf{E}_{ij'})\right)\widetilde{\mathbf{P}}_{ij}. \tag{21}$$

Note that the negative sign of lower bound of the first inequality goes inside of the $\min$ operation while changing it to $\max$ operation. Finally, the matrix level inequality is formulated as follows:

$$(\mathbf{I} - 2\mathbf{D})\widetilde{\mathbf{P}} \leq \mathbf{P} \leq (\mathbf{I} + 2\mathbf{D})\widetilde{\mathbf{P}},$$

where $\mathbf{I}$ is identity matrix. $\mathbf{D}$ is a diagonal matrix whose $i$-th diagonal element is $\mathbf{D}_{ii} = \max_{j,j'}|\mathbf{E}_{ij} - \mathbf{E}_{ij'}|$. We note that the last inequality is even larger/smaller bounds than those of Eq.21 because it takes the maximum value across $j$ and $j'$ simultaneously.

### A.4.4   UPPER BOUNDS OF L-1 AND L-$\infty$ NORMS OF THE MATRIX FORMED $(\mathbf{Z1}softmax(\mathbf{r})^\top)$

By following the property of matrix norm (Sec.A.4.1), the L1 norm of the matrix can be upper bounded as follows:

$$\|\mathbf{Z1}softmax(\mathbf{r})^\top\|_1 \leq \|\mathbf{Z1}\|_1\|softmax(\mathbf{r})^\top\|_1,$$
$$= \|\mathbf{Z1}\|_1,$$

based on the facts that $\|softmax(\mathbf{r})^\top\|_1 = \sum_{i=1}^{T}|softmax(\mathbf{r})_i| = 1$. Similarly, the upper bound of L-$\infty$ norm is as follows:

$$\|\mathbf{Z1}softmax(\mathbf{r})^\top\|_\infty \leq \|\mathbf{Z1}\|_\infty\|softmax(\mathbf{r})^\top\|_\infty,$$
$$\leq \|\mathbf{Z1}\|_\infty,$$

based on the fact that $\|softmax(\mathbf{r})^\top\|_\infty = \max_i|softmax(\mathbf{r})_i| \leq 1$.

### A.4.5   UPPER BOUND OF $\|\mathbf{D1}\|_1\|\mathbf{D1}\|_\infty$

About the L-1 and L-$\infty$ norms of $\mathbf{D1}$, we follow the same definition of Sec.A.4.3 for the matrix $\mathbf{D}$ whose diagonal element is $\mathbf{D}_{ii} = \max_{j,j'}|\mathbf{E}_{ij} - \mathbf{E}_{ij'}|$. Based on the initial condition of Lemma 6 which is related to $\gamma$: $\sqrt{\sum_{i=1}^{T}\max_{j,j'}|\mathbf{E}_{ij} - \mathbf{E}_{ij'}|} \leq \gamma\sqrt{\max_{j,j'}\sum_{i=1}^{T}|\mathbf{E}_{ij} - \mathbf{E}_{ij'}|}$. Then, the multiplication of L-1 and L-$\infty$ norms is derived and upper bounded as follows:

$$\|\mathbf{D1}\|_1\|\mathbf{D1}\|_\infty = \max_{i,j,j'}|\mathbf{E}_{ij} - \mathbf{E}_{ij'}|\sum_{i=1}^{T}\max_{j,j'}|\mathbf{E}_{ij} - \mathbf{E}_{ij'}|,$$

$$\leq 2\max_{i,j}|\mathbf{E}_{ij}|\sum_{i=1}^{T}\max_{j,j'}|\mathbf{E}_{ij} - \mathbf{E}_{ij'}|,$$

$$\leq 2\|\mathbf{E}\|_1\sum_{i=1}^{T}\max_{j,j'}|\mathbf{E}_{ij} - \mathbf{E}_{ij'}|,$$

$$\leq 2\gamma^2\|\mathbf{E}\|_1\left(\max_{j,j'}\sum_{i=1}^{T}|\mathbf{E}_{ij} - \mathbf{E}_{ij'}|\right),$$

$$\leq 2\gamma^2\|\mathbf{E}\|_1\left(2\max_{j}\sum_{i=1}^{T}|\mathbf{E}_{ij}|\right),$$

$$= 8\gamma^2\|\mathbf{E}\|_1\|\mathbf{E}\|_1 = 8\gamma^2\|\mathbf{E}\|_1^2,$$

