# OpenReview forum: "Generalized Probabilistic Attention Mechanism in Transformers"
_ICLR.cc/2025/Conference — Submitted to ICLR 2025_

### Official Review · Reviewer_Jq8M · 2024-11-04

**Soundness:** 3
**Presentation:** 3
**Contribution:** 3
**Rating:** 6
**Confidence:** 4

**Summary:**

Despite the success of Transformers, the standard attention mechanism is associated with two well-known issues: rank-collapse and gradient vanishing. This paper proposed generalized probabilistic attention mechanism (GPAM) to mitigate both the rank-collapse and gradient vanishing issues simultaneously. Evaluation results on real-world datasets demonstrate the effectiveness of GPAM.

**Strengths:**

This paper studies how to alleviate the rank-collapse and gradient vanishing problems in standard attention mechanism, which is an important problem of profound implication in practice. Authors provided a concrete analysis on the intrinsic trade-offs between avoiding rank-collapse and gradient vanishing in standard attention mechanism and showed that GPAM is able to mitigate both problems simultaneously. The theoretical analysis in this paper seems solid and the presentation is clear.

**Weaknesses:**

Evaluation in this work can be significantly improved. Given the prevalent success of Large Language Models (LLMs) on various advanced tasks (e.g., solving math and coding problems), evaluation on only small language models (20M-200M) and classic NLP tasks (e.g., neural machine translation) seems insufficient. More evaluation results on open-sourced LLMs (e.g., Llama-3 or Phi-3 family) and widely used benchmarks (e.g., MMLU or GSM8K) will add great values to this paper.

**Questions:**

Q1. Empirical improvements seem quite marginal according to Table 1. Specifically, a roughly 1% improvement on PPL is not significant enough given that the proposed approach has approximately 1% more parameters (Line 450-451).

Q2. In Line 250, the range for PG should be [1 - \lambda^-, 1 + \lambda^+].

---

> ### Author Response · Authors · 2024-11-24
> **Response to Reviewer Jq8M**
>
> We are glad to read your favorable comments for our work. All those comments encourage us to continue our best effort in this research direction. In this rebuttal, we would like to share our opinion about the weaknesses and questions.
>
> **Weaknesses :**
> *“Evaluation in this work can be significantly improved. Given the prevalent success of Large Language Models (LLMs) on various advanced tasks (e.g., solving math and coding problems), evaluation on only small language models (20M-200M) and classic NLP tasks (e.g., neural machine translation) seems insufficient. More evaluation results on open-sourced LLMs (e.g., Llama-3 or Phi-3 family) and widely used benchmarks (e.g., MMLU or GSM8K) will add great values to this paper.”*
> - We understand the significance of LLM evaluations, so we are trying to do this as future works. In this time, it was hard for us to train LLMs due to our computational budget limit. We guess cost-effective fine-tuning methods (such as LoRA) are insufficient to demonstrate GPAM’s full potential on LLM training, because GPAM adds negative attention part and forms quite a different hyperplane for token representations (refer to ‘4.2 Dynamics of Dual-Attention GPAM’ section) depending on $\lambda$ values. Therefore, to train a GPAM model, it should be from scratch (although it requires massive computational cost). In addition, daGPAM structure requires additional computational cost for attention layers. Therefore, we are now trying to propose a new design for GPAM model to overcome the computational inefficiency bottleneck.
> - Although the current method has inefficient parts, we carefully evaluate that our first step based on daGPAM architecture successively achieved our paper’s main goals, proposing GPAM class and exploring its potential with theoretic evidence and empirical validations including real-world dataset experiments. We expect that this work may motivate future researchers to try better designs of GPAM unveiling its potential in LLMs.
>
>
> **Question1**
> *“Empirical improvements seem quite marginal according to Table 1. Specifically, a roughly 1% improvement on PPL is not significant enough given that the proposed approach has approximately 1% more parameters (Line 450-451).”*
> - Even though the benchmark results appear to be marginal regarding the additional parameters, we would like to carefully argue that GPAM is parameter-efficient, independent development method compared to other parameter scaling method, such as increasing the number of attention head. As in the response to reviewer ‘i3m3’, we provide the comparison of our daGPAM models with the baseline that has +1 additional attention head on top of the baseline. This comparison is motivated by the feature that daGPAM adds one head for the negative part, so it is plausible to raise the question “Does the benefit come from simply scaling up the number of head?”. Below is the result of NMT experiments with additional baseline (+1Head models).
> - IWLST14 En-De (# of parameters, En2De/De2En SacreBLEU) results
>   - PreLN original    64.67M, 28.54/33.90
>   - PreLN +1Head    69.39M, 28.79/34.09
>   - daGPAM (Const) 65.26M, 29.25/34.43
>   - daGPAM (Train) 65.26M, 29.11/34.11
>
> - WMT14 En-De (# of parameters, En2De/De2En SacreBLEU) results
>   - PreLN original    153.84M, 26.40/31.26
>   - PreLN +1Head    158.55M, 26.89/31.55
>   - daGPAM (Const) 155.20M, 26.73/31.43
>   - daGPAM (Train) 155.02M, 27.19/31.45
> - Regarding the number of added parameters due to the +1Head, except the case of WMT14 De2En, daGPAM models achieved enhanced performance with significantly fewer added parameters compared to the '+1Head' models. Additionally, we want to refer to some recent works that show large amount of additional parameters for layer scaling bring only marginal improvements [1] (Table1, vanilla(big) and vanilla(deep) improved roughly 0.5 BLEUs with 2~3 times bigger parameter size compared to vanilla.) and [2] (Table 1 and Table 2, massive number of increased layers bring around 1.0 BLEU improvements.)
>
> **Question2**
> *“In Line 250, the range for PG should be [1 - $\lambda^-$, 1 + $\lambda^+$].”*
> - Those lower and upper bounds are estimated by the two extreme cases, ($P_{ij}^+$=0, $P_{ij}^-$=1) for lower bound and ($P_{ij}^+$=1, $P_{ij}^-$=0) for upper bound. Each case induces $-\lambda^-$ and $1+\lambda^+$, respectively.
>
>
> **References**
>
> [1] Sho Takase and Shun Kiyono. "Lessons on Parameter Sharing across Layers in Transformers." Proceedings of The Fourth Workshop on Simple and Efficient Natural Language Processing (SustaiNLP). 2023.
>
> [2] Wang, Hongyu, et al. "Deepnet: Scaling transformers to 1,000 layers." IEEE Transactions on Pattern Analysis and Machine Intelligence (2024).

---

> > ### Comment · Reviewer_Jq8M · 2024-11-28
> >
> > I want to thank authors for addressing my previous questions. The new experiment results better demonstrate the effectiveness of the proposed approach on small models, though I still feel that this work can be further improved by examining its generalizability to LLMs. Provided that, I will maintain my current overall rating and increase the soundness and confidence scores.

---

> > > ### Author Response · Authors · 2024-11-28
> > >
> > > Thank you for valuable comments :)
> > >
> > > Just for your knowledge, I leave this below information.
> > >
> > >
> > > I didn't upload revised PDF at the previous official comment by mistake..
> > > I just uploaded revised PDF at Nov. 27th 16:00 PM
> > > It contains added appendix A.3.5 'Correlation between Rank-collapse and Performance'.
> > >
> > > Best regards.

---

### Official Review · Reviewer_jf2U · 2024-11-04

**Soundness:** 2
**Presentation:** 3
**Contribution:** 2
**Rating:** 5
**Confidence:** 3

**Summary:**

The paper introduces GPAM, an attention mechanism that allows negative attention scores while maintaining a fixed sum, addressing both rank-collapse and gradient vanishing problems in Transformers. Their proposed dual-attention GPAM (daGPAM), adds a negative attention matrix computation alongside the original one, requiring small additional parameters. The authors show the theoretical advantage of their model and examine their models on language tasks.

**Strengths:**

1. The writing is easy to follow
2. The paper shows quite interesting results gradient vanishing and rank-collapse is hard to solve simultaneously total norm of gradients is maximized can lead to rank-collapse.
3. The authors theoretically show that their method can mitigate both the rank-collapse issue and reduce the the gradient vanishing problem.

**Weaknesses:**

My main concerns are:
1. The improvement of daGPAM is very marginal, and not significant in both Language modeling (on Wikitext-103 and Enwiki8, the improvement is around 0,5 PPL) and machine translation tasks (on IWSLT14 and WMT14, maximum improvement is around 0.7% BLEU score.)
2. Computational cost: the dual-attention structure requires computing two attention matrices instead of one, which is very inefficient. The marginal improvement does not justify this performance-efficiency trade-off.

**Questions:**

1. The optimal $\lambda^{+}$ and $\lambda^{-}$ values seem to vary across different tasks. Can the authors provide an ablation study on choosing different settings of the hyper-params.
2. Can the authors provide the inference time and memory cost of the model?

---

> ### Author Response · Authors · 2024-11-24
> **Response to Reviewer jf2U**
>
> Thank you for your productive reviews. The positive comments regarding the theoretic parts encourage us to continue this research project, aiming for a better future design of GPAM based on this rigorous theoretical foundation. For the comments of weaknesses and questions, we are glad to answer as follows.
>
> **Weaknesses :**
>
> *“The improvement of daGPAM is very marginal, and not significant in both Language modeling (on Wikitext-103 and Enwiki8, the improvement is around 0,5 PPL) and machine translation tasks (on IWSLT14 and WMT14, maximum improvement is around 0.7% BLEU score.)”*
>
> *“Computational cost: the dual-attention structure requires computing two attention matrices instead of one, which is very inefficient. The marginal improvement does not justify this performance-efficiency trade-off.”*
> - As we mentioned in introduction, our main goals of this paper are to propose a new attention class, GPAM, and to explore the potential. We invested most of our effort to the mathematical theories and empirical analyses of the theories. In this sense, dual-attention structure is our first primitive model to simply realize GPAM in actual Transformer to see the first results of GPAM. Although this is the first implementation and the improvement is marginal, the improvement by daGPAM is consistent across the experiments. And, even with such limits, we carefully assess that the demonstrations in this paper will inspire future researchers to experiment with GPAM, employing improved designs that address the limitations of daGPAM, such as computational inefficiency.
>
> **Question1**
> *“The optimal $\lambda^+$ and $\lambda^-$ values seem to vary across different tasks. Can the authors provide an ablation study on choosing different settings of the hyper-params.”*
> - In section ‘A.3.4 Wikitext103 LM Experiments with Varying $\lambda$s’ we conducted the ablation study varying $\lambda$ values. In results, usually, when its fixed sum, (1+ ‘$\lambda^+$’ – ‘$\lambda^-$’), is around 0.5, it achieved the best performance. We conjecture that this result is related to the interpreted dynamics of $\lambda$s in ‘4.2 Dynamics of Dual-Attention GPAM’. We think this finding could be an effective starting point for hyperparameter tuning.
>
> **Question2**
> *“Can the authors provide the inference time and memory cost of the model?“*
> - We provide the requested experimental details.
> - For the Wikitext103 testing with TransformerXL models (64/640 target/memory length setting, default),
>   - 8-layered TransformerXL took ’20.88s’ inference time’ and required ‘12.94GB’ memory in 1x RTX3090 GPU.
>   - 8-layered daGPAM(Const) took ’27.36s’ inference time’ and required ‘15.99GB’ memory in 1x RTX3090 GPU.
>   - 16-layered TransformerXL took ’38.07s inference time’ and required ‘15.09GB’ memory in 1x RTX3090 GPU.
>   - 16-layered daGPAM(Const) took ’49.45s’ inference time’ and required ‘20.67GB’ memory 1x RTX3090 GPU.
> - For the IWSLT14 testing,
>   - PreLN baseline took ‘3m 25s inference time’ and required 9.82GB memory in 1x GTX1080Ti GPU.
>   - PreLN+daGPAM(Const) took ‘3m 52s inference time’ and required 11.15GB memory in 1x GTX1080Ti GPU.
>   - Admin baseline took ‘3m 15s inference time’ and required 9.44GB memory in 1x GTX1080Ti GPU.
>   - Admin+daGPAM(Const) took ‘3m 35s inference time’ and required 10.96GB memory in 1x GTX1080Ti GPU.
> - Even though daGPAM requires more inference time and memory as we conjectured, we expect future works can overcome those inefficiencies (8. Future Works section).

---

> ### Author Response · Authors · 2024-11-28
> **Note for revised PDF**
>
> I didn't upload revised PDF at the previous official comment by mistake..
>
> I just uploaded revised PDF at Nov. 27th 16:00 PM.
>
> It contains added appendix A.3.5 'Correlation between Rank-collapse and Performance'.
>
> Best regards.

---

> > ### Comment · Reviewer_jf2U · 2024-12-02
> > **Reviewer reply**
> >
> > Dear authors,
> >
> > Thank you for your detailed response to my concerns. While I appreciate the time and effort you've put into both the paper and the rebuttal, I must maintain my original position regarding the efficiency-performance trade-off of the proposed daGPAM approach. Additionally, I believe that when introducing a new attention mechanism, it's important to demonstrate clear and meaningful advantages over the standard baseline. I encourage you to explore ways to achieve better or comparable performance with efficient computational requirements in future work.

---

### Official Review · Reviewer_JBvX · 2024-11-06

**Soundness:** 3
**Presentation:** 3
**Contribution:** 2
**Rating:** 5
**Confidence:** 3

**Summary:**

The authors propose daGPAM -- an attention module that computes an additional negative coefficient attention matrix aiming to solve rank collapse and vanishing gradients in transformers.

**Strengths:**

- The paper includes theory on how daGPAM can reduce rank collapse and vanishing gradients. It does so by showing the residual norm of daGPAM and also its gradient is greater than the respective values for vanilla attention (the original values for attention having been computed by [[Dong et al](https://proceedings.mlr.press/v139/dong21a.html)]).
- The paper is well written and easy to follow, and contextualizes relevant work well.

**Weaknesses:**

- I think the downstream experiments for this paper could be fleshed out more. I appreciate that some sort of comparison with alternative attention mechanisms like CoDA is present, but I think only having these comparisons on PTB is insufficient. So many attention variants have been proposed over the years, and none have really taken hold, so I think the empirical bar should be high for current and future variants. Even though this paper shows evidence of reducing rank collapse, rank collapse is ultimately something that matters as it reflects in downstream performance, so I think rank collapse on its own is not enough in the absence of strong empirical results. I think this could be addressed by the authors including more attention variants (CoDA etc) in the LM and NMT experiments.

**Questions:**

- Do the authors believe other attention variants solve the rank collapse / vanishing gradients problem, and can this be experimentally validated?
- How does rank collapse (Figure 4) reflect in downstream performance? It would be interesting to see that a higher degree of rank collapse results in lower downstream results, but without this correlation it is hard to reason about the value of reducing rank collapse.

---

> ### Author Response · Authors · 2024-11-24
> **Response to Reviewer JBvX**
>
> We appreciate your comments for development of our work. Truly, some of the comments have helped us to understand our proposed GPAM in practice. In this rebuttal, we are glad to share our opinion regarding your comments related to weaknesses and questions.
>
> **Weakness1**
> *“I think the downstream experiments for this paper could be fleshed out more. I appreciate that some sort of comparison ~~”*
> - We understand the concerns. We would like to clarify how we selected comparable alternative attention mechanism baselines. The main contributions of this paper are to propose a new class of attention mechanism that especially follows the generalized probability (with two conditions: finite range and fixed sum) and to mitigate the two main problems (rank-collapse and gradient vanishing). For the same purpose, Table 1 aims to demonstrate the importance of generalized probability conditions and the novelty of GPAM as a mitigation method of the two main problems. Therefore, we selected attention variants that are related to our main goals (refer to ‘2. Related Works’ section). As you mentioned, there are a lot of attention variants in these days, but it was not possible to compare our GPAM with all of them.
>
> **Weakness2 :**
> *“Even though this paper shows evidence of reducing rank collapse, rank collapse is ultimately something that matters as it reflects ~~”*
> - We understand your concerns regarding the importance of downstream performance. About the practical benefit by mitigating rank-collapse problem, we cited previous works showing rank-collapse problem’s practical disadvantages [1~3]. From the papers, it is argued that rank-collapse imposes significant bottleneck on training stability and model scaling (e.g., depth-wise). In addition, due to the time and computational budget limits, we concentrated on conducting the correlation analysis that you suggested in the question below (Question2). Please check the response to Question2.
>
> **Question1:**
> *“the authors believe other attention variants solve the rank collapse / vanishing gradients problem, and can this be experimentally validated?”*
> - There are several attention variants that have been proposed to solve the rank-collapse problem and vanishing gradient problem, like enhancing shortcut branch (in residual connection) and identity attention matrix regularization, as we noted in ‘2.1 Rank-Collapse Problem in Transformer” section. Those methods effectively handled rank-collapse problem, but the contextualization effect was weakened because the effect of attention layer was decreased. Thereby, most of those works do not bring downstream performance improvement. On the contrary, GPAM handles the rank-collapse problem with enhancing the effect of main branch (attention layers), so that it brings consistent performance improvement. We analyzed the enhanced influence of attention layers in appendix ‘A.3.3 Influence of Attention Layers’.
> - About the vanishing gradient problem, though some previous attention variants were proposed, they lack performance-wise significance in benchmark experiments (‘2.2 Gradient Vanishing Problem in Transformers” section). On the contrary, our GPAM showed improvements in benchmark experiments.
>
> **Question2**
> *“How does rank collapse (Figure 4) reflect in downstream performance? It would be interesting to see that a higher degree of rank collapse results in lower downstream results, but without this correlation it is hard to reason about the value of reducing rank collapse.”*
> - We appreciate this suggestion for profound understanding of the rank-collapse problem. We added additional correlation analysis in ‘A.3.5 Correlation between Rank-Collapse and Performance’ section of the revised draft (we uploaded during this rebuttal). In short, we computed the correlation between the measured rank-collapse amount in Figure 4 and the matching performances of Wikitext103 in ‘A.3.4 Wikitext103 LM Experiments with Varying $\lambda$s’ section. Among daGPAM models with different $\lambda$ settings, we discovered that ‘residual’ and cosine similarity usually have meaningful correlation with performance. Based on the correlation analysis, we would like to argue that mitigating rank-collapse problem could be helpful for downstream performance.
>
>
> **References**
>
> [1] Dong, Yihe, Jean-Baptiste Cordonnier, and Andreas Loukas. "Attention is not all you need: Pure attention loses rank doubly exponentially with depth." International Conference on Machine Learning. PMLR, 2021.
>
> [2] Noci, Lorenzo, et al. "Signal propagation in transformers: Theoretical perspectives and the role of rank collapse." Advances in Neural Information Processing Systems 35 (2022): 27198-27211.
>
> [3] Noci, Lorenzo, et al. "The shaped transformer: Attention models in the infinite depth-and-width limit." Advances in Neural Information Processing Systems 36 (2024).

---

> ### Author Response · Authors · 2024-11-28
> **Note for revised PDF**
>
> I didn't upload revised PDF at the previous official comment by mistake..
>
> I just uploaded revised PDF at Nov. 27th 16:00 PM.
>
> It contains added appendix A.3.5 'Correlation between Rank-collapse and Performance' which is related to your question.
>
> We are very sorry for this inconvenience,
>
> Best regards.

---

### Official Review · Reviewer_i3m3 · 2024-11-07

**Soundness:** 3
**Presentation:** 4
**Contribution:** 3
**Rating:** 6
**Confidence:** 3

**Summary:**

The paper introduces a new attention mechanism that replaces the attention scores. The attention scores in softmax for each query are a convex sum, so scores must be positive. They argue for using an affine sum which also sums to 1 but allows for negative scores.

They are motivated by rank collapse and gradient vanishing. Rank collapse is mitigated since the negative sums introduce more diversity. They show that gradient vanishing is a related problem.

To validate their method they perform several small scale benchmarks, namely perplexity and translation and find their methods improve against the baselines.

**Strengths:**

The method and presentation are good. They give intuitive, theoretical and experiential justification of the ideas.

The experiment results improve over the baseline.

**Weaknesses:**

The range of evaluations is pretty limited and extra computation is needed.

A benchmark against just increasing the number of attention heads would be useful.

**Questions:**

How does the method work in long context scenarios? Even a simple benchmark like Long Range Arena would be helpful. My intuition feels like there should be a larger improvement.

---

> ### Author Response · Authors · 2024-11-24
> **Response to Reviewer i3m3**
>
> Thank you for your favorable assessments. We are glad to receive your comments and, truly, they are helpful to develop this project in future. In this rebuttal, we would like to discuss the contribution of this paper, and the weaknesses and questions you left for us.
>
> **Weakness1** :
> *“The range of evaluations is pretty limited and extra computation is needed.”*
> - We understand the comment regarding the range of evaluations based on language modeling (LM) and neural machine translation (NMT). However, LM and NMT are classical tasks in recent NLP domain. Moreover, LM (and NMT as a conditional version of LM) is definitely the fundamental pre-training strategy that can affect a lot of downstream tasks, such as question-answering and summarization. Therefore, we expect that the improvement in LM and NMT is crucial and can be reproduced in other tasks.
> - As we depicted in ‘8. Future Works’ section, we understand that current daGPAM model requires extra computation. Nevertheless, we would like to argue that the results of daGPAM show the main goal of our paper, which is to propose GPAM and explore GPAM’s potential (1. Introduction section). As we mentioned in ‘8. Future Works’ section, we expect future works can overcome the computational inefficiency of this primitive model, daGPAM, while preserving (or enhancing) the concept of GPAM with better models.
>
> **Weakness2** :
> *“A benchmark against just increasing the number of attention heads would be useful.”*
> - We appreciate this suggestion for further comparison. During the development process of this project, we had trained PreLN benchmark models with +1 head for IWSLT14 and WMT14 tasks. Due to the page limit, we could not report this comparison, but we are glad to provide this comparison during this rebuttal as follows.
>
> - IWLST14 En-De (# of parameters, En2De/De2En SacreBLEU) results
>   - PreLN original    64.67M, 28.54/33.90
>   - PreLN +1Head    69.39M, 28.79/34.09
>   - daGPAM (Const) 65.26M, 29.25/34.43
>   - daGPAM (Train) 65.26M, 29.11/34.11
>
> - WMT14 En-De (# of parameters, En2De/De2En SacreBLEU) results
>   - PreLN original    153.84M, 26.40/31.26
>   - PreLN +1Head    158.55M, 26.89/31.55
>   - daGPAM (Const) 155.20M, 26.73/31.43
>   - daGPAM (Train) 155.02M, 27.19/31.45
>
> - Please note that daGPAM’s additional negative head shares most of parameters with the original positive head, so it does not increase parameters significantly compared to +1Head baselines. Interestingly, except WMT14 De2En experiment, we found that daGPAM model still outperforms +1Head baselines. We believe that the additional negative head in daGPAM can deliver an independent benefit in terms of performance, in a parameter-efficient manner.
>
>
> **Question:**
> *“How does the method work in long context scenarios? Even a simple benchmark like Long Range Arena would be helpful. My intuition feels like there should be a larger improvement.”*
> -  We appreciate this suggestion. As you mentioned, we also expect additional improvement on long range contexts, because the rank-collapse problem might be harder in longer contexts. However, due to the limit of our computational budget, it is hard to train daGPAM on longer contexts than the current setting (TransformerXL with 150/640 (train/test) token lengths). We plan to conduct this experiment once we have access to a more powerful server, or once a more computationally efficient design for GPAM is developed.

---

> ### Author Response · Authors · 2024-11-28
> **Note for revised PDF**
>
> I didn't upload revised PDF at the previous official comment by mistake..
>
> I just uploaded revised PDF at Nov. 27th 16:00 PM.
>
> It contains added appendix A.3.5 'Correlation between Rank-collapse and Performance'.
>
> Best regards.

---

### Meta-Review · Area_Chair_nMdr · 2024-12-20

**Metareview:**

This work introduces daGPAM, an architecture update addressing two problems in transformers : rank collapse and vanishing gradients. Reviewers enjoyed the theoretical contributions, but the empirical validation of the proposed approach is limited. The improvements observed on standard benchmarks are marginal, and the paper lacks a comprehensive comparison with other alternative architectures. Overall, the paper falls short of the acceptance threshold.

**Additional Comments On Reviewer Discussion:**

This paper received borderline ratings. The authors submitted a rebuttal and 2/4 reviewers acknowledged it but didn't change their scores.

---

### Decision · Program_Chairs · 2025-01-22

Reject